

# High-resolution marine flood modelling with coupled overflow and overtopping processes: framing the hazard based on historical and statistical approaches.

Alexandre Nicolae Lerma[1], Thomas Bulteau[2], Sylvain Elineau[1,3], François Paris[2], Paul Durand[3], Brice Anselme[4], Rodrigo Pedreros[1]

[1]BRGM (French Geological Survey), Risks and Prevention Division - Coastal Risks and Climate Change Unit, Orléans, France
[2]BRGM (French Geological Survey), Regional Direction Nouvelle-Aquitaine - Pessac, France
[3]LGP / Université Paris 1, UMR 8591
[4]PRODIG / Université Paris 1, UMR 8586

*Correspondence to*: Alexandre Nicolae Lerma (a.nicolaelerma@brgm.fr)

**Abstract.**

A modelling chain was implemented in order to propose a realistic appraisal of the risk in coastal areas affected as well by overflowing as overtopping processes. Simulations are performed through a nested downscaling strategy from regional to local scale at high spatial resolution with explicit buildings, urban structures such as sea front walls and hydraulic structures liable to affect the propagation of water in urban areas. Validation of the model performance is based on hard and soft available data analysis and conversion of qualitative to quantitative information to reconstruct the area affected by flooding and the succession of events during two recent storms. Two joint probability statistical approaches (joint exceedance probability and environmental contour) are used to define 100 years off-shore conditions scenarios and to investigate the flood response to each scenario in term of: (1) maximum spatial extent of flooded areas, (2) volumes of water propagation inland and (3) water level in flooded areas. Scenarios of sea level rise are also considerate in order to evaluate the potential hazard evolution. Our simulations show that for a maximising 100-year hazard scenario, for the municipality as a whole, 38% of the zones are prone to overflow flooding and 62% to flooding by propagation of overtopping water volume along the seafront. Results also reveal that for the two kind of statistic scenarios a difference of about 5% in the forcing conditions (water level, wave height and period) can produce significant differences responses in terms of flooding like +13.5% of water volumes propagating inland or +11.3% of affected surfaces. In some areas, flood response appears to be very sensible to the scenario chosen with differences of 0.3 to 0.5 m in water level. The approach developed enable to bracket the 100-year hazard and to characterise spatially the robustness or the uncertainty over the results. Considering a 100-year scenario with mean sea level rise (0.6 m), hazard characteristics are dramatically changed with an evolution of the overtopping/overflowing process ratio and an increase of 384% in volumes of water propagating inland and 247% in flooded surfaces.

**Key word : flood hazard, numerical modelling, joint probability, sea level rise, Mediterranean sea**



## 1 Introduction

Awareness of the increasing vulnerability of coastal cities to storms and expected effects of global warming lead to more and more studies focusing on the risks of coastal flooding in coastal lying areas. These studies often conclude that even a relatively slight rise of mean sea level will, in areas that are not actually exposed or where the risk is currently manageable, trigger more frequent hazard and potential disastrous consequences (Hunter, 2012; Tebaldi et al., 2012). On many low-lying coastlines, a "tipping point" is likely to be reached with a mean rise in sea level of 0.5 m (Sweet and Park, 2014).

Apart from a failure in flood defences, coastal flooding is mainly triggered in two ways. Overflow flooding occurs when the static sea level rises above the level of the natural terrain or flood defence. Overtopping occurs when a combination of high sea level and breaking waves cause successive sheets of seawater to sweep over the seafront.

Coastal flooding risks are usually defined by the intensity of flooding (spatial extent, water height, flow speed, etc. or a combination of these parameters) associated with the probability of occurrence, usually defined as the "return period".

Low-lying areas exposed to waves can be flooded successively or simultaneously by overflowing and overtopping along the same coastline. In these conditions, risk mapping using simple methods (cross-referencing topography and sea level), decametric DTM resolutions or without taking buildings into account will not produce adequate or realistic details of the risks to urban areas. High-resolution numerical modelling has therefore become the preferred approach to characterise flooding risks in the most exposed and vulnerable sites (e.g. Guimarães et al., 2015; Leroy et al., 2015; Gallien, 2016).

Models of overflow flooding are now relatively accurate and usually based on well proven physical and numerical methods that have been applied to river, coastal and estuarine contexts and are capable of representing well the extent of flooding. These include the semi-static method (e.g. Breilh et al., 2013), cellular automata (e.g. Dearing et al., 2006; Hawick, 2014), and hydrodynamic modelling (Martinelli et al., 2010; Gallien et al., 2011; Smith et al., 2012; Fortunato et al., 2013).

Models simulating overtopping are much more recent and still require substantial research developments (Hubbard and Dodd, 2002; Gallien, 2016). In the last few years, several process-based models have been developed and applied to address coastal flooding risks: VOF model (Tomas et al., 2014), Boussinesq model (Lynett et al., 2010; Andrade et al., 2013), NSWL model (Suzuki et al., 2011; Guimarães et al., 2015; Leroy et al. 2015), and Non-Hydrostatic Phase-Averaged Model (Smith et al., 2012; Gallien, 2016). These, and especially the SWASH model, are able to reproduce the dynamics of wave surges and overtopping to an appropriate degree of reliability for coastal flooding studies (Suzuki et al., 2011). However, questions remain as to the order of magnitude of overtopping volumes, whether estimated empirically (Laudier et al., 2011; Gallien et al., 2014) or by digital modelling (Smith et al. 2012; Gallien 2016). In some cases, the estimated orders of magnitude can vary by as much as a factor of 10 (Lynett et al., 2010). These estimated uncertainties as to the reproduction of overtopping volumes can also be attributed to the inadequacy of validation data, which are often qualitative and partial (Battjes and Gerritsen, 2002; Poulter and Halpin, 2008; Reeve et al., 2008; Anselme et al., 2011; Gallien et al., 2012).

As yet, only a few studies have attempted to couple flooding by overflowing and overtopping (Gallien et al., 2014; Stansby et al., 2013; Le Roy et al., 2015; Gallien, 2016). Le Roy et al. (2015) have attempted to integrate the spatial and temporal



variability of overtopping by simulating overtopping in 2D. This type of model requires a high spatial resolution (less than 2 m), computing resources and time required to cover sites over several kilometers in extent are still prohibitive. The solution used in our study was to link several models into a chain in order to reproduce, on the one hand, variations in mean sea level, including tides, storm surges and wave setup, by coupling a hydrodynamic model (MARS, Lazure and Dumas, 2007) with a

spectral wave model (SWAN, Booij et al., 1999), and on the other hand, to assess runup and overtopping volumes at the seafront an NLSW model (SWASH, Zijlema et al., 2011). The chained modelling enabled us to model the different coastal flooding process (overflowing or overtopping) and consequences at a high resolution over a spatial extent of several kilometers.

Coastal flooding risks are also usually associated with a return period (i.e. probability of occurrence). The classic approach

recommended in several EU (WFD, APSFR) and national directives (PPRL in France) involves running several scenarios with different probabilities of occurrence (10, 50 or 100 years) plus several scenarios for the same return period. Numerous studies have focused on multivariate extreme value analyses of interdependent meteorological and marine variables (for a review, see Jonathan and Ewans, 2013 and Monbet, 2007). The complexity of a multivariate extreme value analysis is due to the inadequacy of current knowledges on the interdependence of variables in the tail of the multivariate distribution. An

estimation of the tail behaviour is therefore required. Among the existing statistical models used to represent dependence in the tail of the distribution, the semi-parametric model for conditional extreme values first derived by Heffernan and Tawn (2004) (see, among others, Zheng et al., 2013; Wyncoll and Gouldby, 2015; Gouldby et al., 2014) is increasingly used in hydrological, coastal and ocean engineering applications. This model overcomes some of the limitations of classic approaches, such as copulas (e.g. De Michele et al., 2007, Salvadori et al., 2011, Wahl et al., 2012, Wahl et al., 2013,

Hawkes et al., 2002, Hawkes et al., 2008, Masina et al., 2015); in particular, it does not require any assumption about the dependence structure and can be easily extended to larger dimensions (typically larger than 2). While the return period in a univariate case is clear and unambiguous, it is much less so when two variables or more are considered together (Salvadori et al., 2011). Moreover, in most risk studies involving several variables, it is considered in the selected scenarios that the return period defined on the basis of the input variables corresponds to the return period of the system response, i.e. flooding in our

case. However, the system response is often complex, and whenever a problem addressed has more than one dimension, the one-to-one relationship is rarely valid (Idier et al., 2013). In the related field of structural engineering (design of coastal defences, offshore renewable energy systems, etc.), it is usual to refer to reliability methods (Jonathan and Ewans, 2013; Winterstein et al., 1993; Huseby et al., 2013; Huseby et al., 2015). Reliability methods focus on estimating extreme system responses rather than on the combinations of extreme events that produce these responses. In this case, the scenarios are

chosen to obtain a system response with the return period under consideration. Such a method is rarely used to assess coastal flooding risks. In this paper, it will be compared with the more classic method, where in choosing the scenarios it is assumed that the return period defined from the input variables corresponds to the return period of the system response, in order to analyse the differences that arise from the methods used to define the scenarios.



This study therefore has two aims: the first is to use tools able to produce realistic representations of flooding by comparing the simulations to existing qualitative and quantitative data from past events and differentiating between the processes causing the flooding. The second aim is to assess, using the same tools, the risk of coastal flooding with a low probability of occurrence according to different statistical methods for defining scenarios with the current mean sea level and with the

mean sea level rise expected during the 21st century.

In the first section, we describe the study site and the methods implemented. The second section presents the results. In the third section, we discuss the methods used and the results, before presenting our conclusions.

## 2 Flood modelling: data and methods

### 2.1 Study site

Like many other beach resorts in the Languedoc Roussillon area (SW France), our study site is highly exposed to coastal flooding risks. The municipality of Leucate lies on the western side of the Gulf of Lion, with the Mediterranean on its eastern side and the Salses-Leucate lagoon to the west (Fig 1). The coast has a microtidal regime (0.2 to 0.4 m) with low-energy mean wave conditions at Hs = 0.67 m; Tp = 4.6 s (observation period: 12/2006 - 03/2013) and prevailing winds from north-east (Fig 1).

Figure1

The circulation pattern of winter storms, characterised by significant storm surges (0.6 m to 1 m) and very intense wave conditions from the east-south-east (over 6 m in height with peak periods of 10 to 12 s), is damaging the seafront and

causing recurrent flooding in different parts of the town (seafront, harbour and lagoon passes, Fig 2).

Figure2

On the site the coastal flooding risk is mainly due to two aspects : the hazard is related to a general low lying topography,

particularly in the inner part of the lido were exchanges of water between sea and lagoon are constrained. Second, the vulnerability is high due to a massive urbanization and the fact that some neighbourhoods have been built directly onto the top of the beach, on the foredune (Fig. 2). For analysis, three area were distinguished in the study site: Leucate Plage (Zone A), the naturist village (Zone B) and Port Leucate (Zone C). We also used several beach profiles along the coastline illustrating the spatial variability of sea front in topography and main structures (Wall, Built Sea Front, Back Beach Low)

(Fig. 2).



## 2.2 Forcing data

We used different sources of wave data for the study (Table 1): (i) observation data from the Candhis 01101 buoy (hourly intervals over a discontinuous period from 2007 to 2015) for local wave parameters used as benchmarks for sea-state modelling and for the statistical study; (ii) data from IFREMER MEDNORD, code WWIII, 0.5°x 0.5° resolution (IOWAGA), used as forcing data to model past events; (iii) a time series extracted from retrospective simulations (NOAA-CFSR-med_10m forcing) with the SWAN model (Booij et al., 1999) on a Mediterranean grid (42°N-44°N/2°E-8°E) with a resolution of about 1 km (Stépanian et al., 2014). These data (abbreviated here as GuLWa for Gulf of Lion Wave data base), covering a 31-year period (1979-2009) at hourly intervals, were collected at the Candhis 01101 buoy location and used for the statistical analysis, and especially to adjust the marginal distribution of Hs peaks (cf. section 2.4).

Table 1

The tide gauge closest to the study site is located at Port la Nouvelle (SHOM/CR LRO) about 15 km to the north of the site (Fig. 1), but could only provide recent data (2013-2015). These data were used to reproduce an event in the recent past (November 2014). For our analysis of earlier events and to conduct the statistical analysis, the SHOM/CR LRO tide gauge offshore from Sète, 80 km to the north (Fig 1), was the only one able to provide sufficient data. The wind data used for the study are from the Leucate semaphore (Météo France data).

## 2.3 Topo-bathymetric data, built structures, surface roughness

High-resolution modelling of coastal flooding risks requires a finely detailed representation of the bathymetry and topography. Significant data collecting efforts were needed to produce an accurate representation of the study sector, including the land-sea continuum, the land areas, the lagoon and passes, the harbour and the nearshore and offshore areas (Fig 2, Table 2). All data are presented in French national topographic reference (NGF).

Table 2

Numerous studies of the land area have shown that urban structures such as walls and banks can have a determining role in the dynamics of flooding (water flow and extent) and therefore need to be included in the representations of urban environments produced by digital models (Bernatchez et al., 2011; Brown et al., 2007; Fewtrell et al., 2008; Gallegos et al., 2009; Gallien et al., 2011; Mignotet al., 2006; Poulter and Halpin, 2008; Néelz et al., 2006). To represent these structures, altimetric data from LIDAR grids (DEM, DTM at 1m resolution) are essential core data. To represent buildings, the necessary data were extracted from the Litto3D DEM via cross-referencing with the "built-up" layer (undifferentiated,





industrial and outstanding buildings) from the IGN Topo database. Only areas >20 m² were taken into account (except isolated or apparently precarious structures) and areas of 20-50 m² for isolated structures (more than 100 m away from another). This "built-up" layer was then draped over the Litto3D DTM. This enabled us to include only buildings likely to obstruct water flow and to filter out any vegetation or noise in the raw model.

The horizontal resolution (1 m) of the core data and their degree of vertical accuracy, usually ± 20 cm, were not sufficient to represent some structural elements that are fundamental in constraining and reproducing inland flows propagation. Some localised retouching should therefore be considered (Poulter and Halpin, 2008; Smith et al., 2012) to incorporate these details into the model.

A ground survey was carried out in June 2015 to set up control points for the different data sources and to make an inventory
of elements that were not detectable or only represented discontinuously in the available LIDAR dataset. The topographic elevations and functional hydraulic characteristics of coastal retaining walls and hydraulic structures liable to affect the propagation of water masses were measured and incorporated so that the DTM grid cells concerned are automatically enhanced by the D-GPS survey values.

Based on these data, two topo-bathymetric models at different spatial resolutions were built up (Fig. 2), one at 20 m
resolution (Rank 0: 825 x 827 grid cells) covering the entire stretch of the Salses-Leucate lagoon, and one at 5 m resolution (Rank 1: 606 x 1576 grid cells), covering the Leucate municipality (Port-Leucate and Leucate-Plage). Cross-shore profiles at 1 m resolution were also used to model overtopping along the sea front of the study area.

To ensure that flows are properly represented, it is necessary to consider the land use is represented in models. Land use is incorporated into the models through a variable friction coefficient that depends on the soil type and the type of urbanisation
according to density (Leroy et al., 2015; Bunya et al., 2010, see Table 3).

Table 3

In this study, a spatialised representation of terrain roughness was obtained from a synthetic land use classification based on
2006 Corine Land Cover data. However, the Corine Land Cover data are to the scale of 1:100 000, which is not suited to the scale of our study. The data were therefore re-sampled and their footprint modified from ortho-photographs to generate a suitable 20m-resolution roughness map. The values used to characterise roughness are those recommended by different sources as applicable to studies conducted in the marine and coastal domains (Bunya et al. 2010; Goutx and Ladreyt, 2001). Specific processing was carried out to represent roads, which are zones where water circulates easily due to the lack of
obstacles and the nature of road materials (concrete and tarmac).





## 2.4 Flood modelling chain

Modelling of coastal flooding involved running several chained models: the MARS-2DH hydrodynamic model (Lazure and Dumas, 2007), the SWAN wave spectre model (Booij et al., 1999) and the SWASH non-linear shallow water model (NLSW) (Zijlema et al., 2011) (Fig. 3).

5                                                          Figure 3

We used the MARS computing code (Lazure and Dumas, 2007) to assess the regional hydrodynamics based on tidal components and meteorological data. The processes represented by the model are associated with long wavelengths only (tides and storm surges). We used the 2DH version of the model, which resolves the Saint-Venant equations that govern

horizontal free-surface flows in two dimensions, after vertical integration of the Navier Stokes equations.

When linked to the SWAN wave model, the MARS-2DH model includes short wavelength interactions between waves, sea level and currents (swells and wind sea), mainly in the coastal zone, and can thus calculate the additional water height resulting from wave setup. MARS-2DH thus calculates the speed and direction of currents, averaged to the vertical, and water heights, according to the limit conditions imposed at the edges of the computed domain (boundaries) and the

meteorological forcing applied at each node in the model.

In sectors prone to coastal flooding by overtopping across the seafront, the overtopping volumes are estimated via 1D modelling with the SWASH model (Zijlema et al., 2011). The SWASH model is a time domain model for simulating non-hydrostatic, free-surface and rotational flows. The governing equations are the shallow water equations including a non-hydrostatic pressure term. This model, whose performance in reproducing overtopping volumes was demonstrated by Suzuki

et al. (2011), is used here to estimate runup and water volumes likely to overtop seafront walls according to their geometry. The water volumes along the length of the zone concerned are reinjected into the calculation for flooding behind the seafront and seawalls, in order to reproduce the inland propagation of overtopping volumes.

After completing the simulations, the coastal flooding risk is defined by the intensity of submersion, described here by three types of information: the maximum spatial extent of flooded areas (written as $S_{flood}$), the volumes of water reaching inland

(written as $V_{flood}$) and the spatially variable height of the floodwater (written as $H_{flood}$).

## 2.5 Exploiting historical data: storm conditions and flooded area

### 2.5.1. Water level and wave conditions

Two storm events in March 2013 and November 2014 were analysed in order to assess the performance of the linked models

in reproducing the observed flooding events. These two events were characterised by different marine characteristics and consequences in terms of coastal flooding. The data available to characterise the storm conditions were describes in Table 1.



For the November 2014 storm, the sea level data are from the Port-la-Nouvelle tide gauge. Because no data from this station were available for the March 2013 storm, the water level forcing data are from the Sète tide gauge. However, our analysis of the periods common to both tide gauges covering this stretch of the Gulf of Lion coast (Sète, Port la Nouvelle, Banyuls (Fig. 1) shows that for the events studied, the associated storm surges were fairly uniform along the Languedoc-Roussillon coast.

The peak water level of the March 2013 storm was a 0.15 m difference at the Sète (0. 97 m/NGF) and Port Vendres (1.12 m/NGF) tide gauge, located respectively at 80 km Nord East and 40 km south from the study site.

The wave data used to reproduce these events were extracted from the IFREMER MEDNORD model at the limits of the domain investigated (Rank 0), (Fig. 4). The quality of the reproduction of wave conditions in the domain studied was cross-checked with data from the Leucate buoy, with very good fitting (not show here).

### 2.5.2 Flood observations and field measurements

To help characterise the quality of coastal flooding modelling, we use several source of available information, from "hard" to "soft" data (Smith et al., 2012). Information was compiled from a wide range of sources, "hard" data from photographs, reports from technical departments, and "soft" data from press, interviews and eyewitness accounts. This material enabled us

to reconstruct the zones affected by flooding and the succession of events during the storms. Although often qualitative, these observations allowed to estimate water levels reached locally, based on urban landmarks (pavements, walls, jetties, etc). Each observation point was then cross-checked against LIDAR and/or DGPS measurements to produce quantitative information "hard data" from the qualitative validation "soft data" (Table 4). The limits reached by floodwaters in the worst affected sectors were also mapped with the help of the municipal agents who worked on the ground during the storms.

As no local tide gauge data on water levels in the harbour were available, we were able to extract validating material from these documents to assess the quality of the model's reproduction of water levels and flooding at different points across the study area (Fig. 4, Table 4).

Figure 4

Table 4

During the 2013 storm, a breach observed in the seawall to the north of the municipality caused flooding in a large area of the village. The breach, 15 m in length, occurred because the seawall had not been designed to withstand the full weight of the water accumulating through wave action. Based on the limits and heights of the floodwater described by the Leucate municipal agents and inhabitants, and observed from photographs, we were able to reconstruct the extent and height of the

flooding in the village of Leucate-Plage (Fig. 4). The volume of water that flooded the village as a result of the breach was estimated (by cross-referencing topographic and water level data) at a minimum of 37 000 m$^3$ (this figure is taken as a minimum because several instances of overtopping were observed in the non-urbanised zone to the south, which are hard to quantify) (Fig. 4, Table 4).





Simulations were run to reproduce the extent of the flooded zones and the water heights that affected the sector. The modelling chain was not able to reproduce the breach in the seawall or the speeds of the resulting water flow. The aim here was to reproduce the consequences of a breach in terms of flooding (extent of the flooded zone and height of the floodwater) by propagating a certain volume of water from the breach zone. Two methods were used to quantify this water volume.

The first involved the flooding model only. Locally, the breach was simulated by applying the laws of hydraulic thresholds (flooded and dewatered conditions) to calculate the upstream to downstream flows from the breach. This incorporated the breach into the grid as a hydraulic singularity without modifying the topography in the model. The geometry of the breach was simplified into a rectangle with a fixed width and a variable threshold over time.

The second method involved simulating the breaking waves by running the SWASH model with a profile facing the breach
zone. During the simulation, the seawall was erased at the point in time when the breach occurred. The water volumes coming through the breach were then injected into the propagation model. With this approach, the accelerating speeds and water volumes likely to flow through the breach are not taken into account.

The results obtained in terms of flooding were compared to available information from the ground on the extent of flooded areas (written as $S_{flood}$) and water volumes inland (written as $V_{flood}$) as estimated via GIS methods.

# 3 Statistical approach

The aim here was to produce scenarios for offshore marine conditions with a low probability of occurrence that propagate to the shoreline and then inland. To do so, a multivariate analysis of extreme values (waves and water levels) was conducted to artificially enlarge the dataset so that scenarios could be selected from the results of two different methods, one based on the
return period of offshore marine conditions ("joint exceedance contours") and the other on the return period of the hazard ("environmental contours").

## 3.1 Multivariate extreme values method

The interdependence of offshore forcing variables is modelled here using the semi-parametric approach developed by
Heffernan & Tawn (2004). This approach extrapolates the joint probability density of the offshore marine variables (Hs, SWL) in the extreme values domain by considering the structure of dependency between the variables. For detailed description of the method, readers may consult, in particular: Heffernan and Tawn, 2004; Gouldby et al., 2014; Wyncoll and Gouldby, 2015. Here, we provide an outline of the main steps followed to implement our case study:

Data preparation

The available data from Sète make up a continuous series for the 1996-2015 period, corresponding to 16.4 actual years. For the statistical analysis, the long-term linear trend in sea-level rise was eliminated and the values attached to the official mean





sea level for Port La Nouvelle: 0.59 m above Z.H. (French chart datum) (SHOM, 2014). The wave data used are from the CANDHIS 01101 buoy for 2007-2015. The simulated data (Stépanian et al., 2014), co-localised at the buoy and covering the 1979-2009 period, were also used to adjust the marginal Hs distribution (see below).

Concerning storm dynamics in the Gulf of Lion, we decided to select the maximum Hs values per 3-day block, with a
minimum of 1.5 days between peaks to make sure of their independence. For each peak Hs value, the SWL maximum was then sought within a 12-hour window with the Hs peak at its centre. Each Hs value was associated with the corresponding peak period Tp and peak direction Dp. Several quadruplets (Hs, Tp, Dp, SWL) were thus selected, corresponding to about 6 years of common data covering 111 events/year on average. Given the exposure of the coastline and the wave direction during storms, only waves from the 60°-210° sector were kept for the analysis.

The Tp and Dp variables are treated as covariables of Hs: as the peak period is highly dependent on Hs and is not an amplitude variable like Hs and SWL, it is considered here as a covariable, as is the peak direction Dp.

Marginal distributions

Adjustment of marginal probability distributions $F_i$ for each variable $X_i$: when a properly selected high threshold $u_i$ is
exceeded, this is modelled via a Generalised Pareto Distribution (GPD). Below this threshold, the empirical distribution $\tilde{F}_i$ of each variable is used:

$$F_i(x) = \begin{cases} \tilde{F}_i(x) & x \le u_i \\ 1 - \left(1 - \tilde{F}_i(u_i)\right)\left[1 + \frac{\xi_i(x-u_i)}{\sigma_i}\right]_+^{-1/\xi_i} & x > u_i \end{cases} \tag{1}$$

Where $\xi_i$ and $\sigma_i > 0$ respectively are the GPD form and scale parameters and $z_+$ for $z \in \mathbb{R}$ is defined as $z_+ = \max(z, 0)$.

The Languedoc coastline has a microtidal regime that does not warrant the use of indirect methods, i.e., separating the deterministic signal (tide) from the random signal (storm surge), to calculate extreme water levels (Haigh et al., 2010). A direct method was therefore employed to analyse the extreme signal values.

The marginal SWL distribution was calculated from the truncated Sète tide gauge series (see above), i.e. covering about 16.4 years. The time series was first re-sampled in the same way as to make up the sample of (Hs,SWL) pairs over the common
time span, i.e. by taking the maximum water level per 3-day block, then a statistical threshold was chosen beyond which the GPD is adjusted to the data. The threshold was chosen by applying several techniques based on a visual appreciation of quantile-quantile graphs, "mean residual life plots", "modified scale and shape parameters plots" and statistical tests such as the χ² test and the Kolmogorov-Smirnov test (Coles, 2001; Nicolae Lerma et al., 2015). The best fit among 3 methods for estimating GPD parameters ($\xi$ and $\sigma$), namely maximum likelihood (ML), method of moments (MOM) or probability
weighted moments (PWM), was then chosen on the basis of visual and statistical tests (Nicolae Lerma et al., 2015). For the SWL variable, the best fit was achieved with the MOM beyond the 0.96 m Z.H. threshold (Figure 5a).



Figure 5

The wave observation data (Candhis) cover only 7 years, discontinuously, which is too short a period to extrapolate the distribution of probability in the extreme range and to consider long return periods (typically 100 years).

This is a classic problem for any analysis of extreme values from observation data. In order to extrapolate probability distributions in the extreme range, the amount of data has to be sufficient to reduce statistical uncertainties to a reasonable level and thus produce meaningful results. When observation data cannot be used or are unavailable, a possible alternative is to use model output (re-analyses). However, in this case, errors attributable to the model (e.g. lack of precision in spatio-temporal resolution, bathymetry or forcing data) are transferred to the statistical analysis and generate uncertainties as to the results (Caires and Sterl, 2005; Mínguez et al., 2012).

Bulteau et al. (2015) developed a method (called HIBEVA for Historical Information in Bayesian Extreme Value Analysis) for using historical data from archives to analyse extreme water level values. The flexibility and overall Bayesian framework of HIBEVA justify its use in this study to estimate the marginal probability distribution of significant wave heights via a combination of observation data and model output. The observation data (Candhis) are treated as systematic data and the modelled data (GuLWa) are treated as uncertain historical information. We therefore only used the GuLWa data for 1979-2006 (i.e. before the Candhis data came on line).

To estimate the uncertainties relating to the model output data, a comparison (not presented here) was made between the two datasets over the common period from 2007 to 2009. From this we deduced a working hypothesis: for the 1979-2006 period, the true Hs peak values fall within an interval I = [peak Hs from GuLWa - 0.15m; peak Hs from GuLWa + 1.60m].

Similarly to the treatment of water levels, the time series (observation data and model output) were first re-sampled taking the maximum Hs per 3-day block, then a statistical $u_s$ threshold was chosen (based on observation data only) beyond which the GPD is adjusted to the data using the HIBEVA method. This also requires a "historical perception threshold". In this case, the threshold was set at $u_s + 0.15m$ so that the lower limit of the interval I would be equal to at least $u_s$.

Figure 5b shows the results of applying the HIBEVA method for Hs. The $u_s$ threshold is set at 2 m. The chosen GPD parameters (solid red curve) correspond to the mode of the *a posteriori* joint probability distribution of GPD parameters (see Bulteau et al., 2015 for details).

By combining GuLWa and Candhis data, the actual duration of observations for statistical wave analysis can be extended from 7 years (Candhis data only) to 35 years. With Candhis data only, the maximum return period that could be considered was around 30 years (about 4 times the duration of observations, Pugh, 2004). The maximum return period now is around 140 years.

Fitting the dependency model in the Gumbel space





Original variable $X_i$ are transformed into common standard Gumbel margins $Y_i$ using the standard probability integral transform. Then, if $\boldsymbol{Y_{-i}}$ is the vector for all variables except $Y_i$, the non-linear multivariate regression model is as follows :

$$\boldsymbol{Y_{-i}}|Y_i = \boldsymbol{a}Y_i + Y_i{}^{\boldsymbol{b}}\boldsymbol{W} \qquad \text{for } Y_i > \nu \text{ and } Y_i > \boldsymbol{Y_{-i}} \text{ (i.e. } Y_i \text{ being maximum)} \qquad (3)$$

Where **a** and **b** are parameters vectors (one value per parameter for each pair of variables), $\nu$ a threshold to be defined and $\boldsymbol{W}$
a vector of residuals. The model is adjusted using the maximum likelihood method on the assumption that the residuals $\boldsymbol{W}$
are Gaussian with a mean and variance to be calculated.

For our case study, the threshold selected for $\nu$ Eq3 was 0.75 (expressed as a probability of non-exceedance) using the diagnostic tools described in Heffernan and Tawn (2004).

Monte Carlo simulation

The next step was a Monte Carlo simulation to artificially generate **Y**, keeping to the original proportion of events where each $Y_i$ is a maximum.

For our case study, we simulated a fictitious 10 000 year period. These 10 000 years should not be construed as a prediction or forecast for the future, but they are representative of currently available data. Figure 6 shows the results of the simulation.

To finish, the Gumbel variables $Y_i$ are transformed back into the original space. The final output is a large sample of
artificial offshore sea conditions where at least one variable is extreme (exceeding a defined threshold) and which respects both the individual marginal distributions and the structure of dependence between variables.

Figure 6

## 3.2 Defining the multivariate scenarios

### 3.2.1 Joint exceedance contour

Once the sample of offshore marine data has been artificially enlarged, scenarios for the return period T considered (here, T=100 years) were selected for propagation. A commonly used approach in the field of coastal risks involves choosing combinations of forcing factors with a joint exceedance return period equal to T. The idea is then to calculate the joint exceedance contour (written here as *jec*), i.e. the contour (x,y) within the space (SWL,Hs) whereby the joint exceedance probability $P(SWL > x, H_s > y)$ is constant (and equal to the probability associated with T) at every point around the
contour (see Fig. 7):



$$P(SWL > x, Hs > y) = \frac{1}{\lambda T} \tag{4}$$

Where $\lambda$ is the average number of events per year (111 in our case).

We then need to find the maximum response Z (e.g. flooded area, maximum water height inland) along the contour (Hawkes et al., 2002). Practically speaking, this means separating the contour into a number of discrete combinations (SWL, Hs) that will all propagate inland (Fig. 7). The maximum response from these propagations is then associated with a return period T, and written as $z_T^{jec}$.

As underlined previously, this approach rests on the assumption that the return period of the response is equal to the return period of joint exceedance of the input variables. In reality, the joint exceedance probability for the response variable is an underestimation (Hawkes et al., 2002; Idier et al., 2013; Bulteau et al., in prep). The reason for this is simply that combinations which do not belong to the space $(SWL > x, H_s > y)$ can still produce values for the response variable Z in excess of $z_T$.

### 3.2.2 Environmental contour

A second approach involves using environmental contours (written here as *enc*), which are commonly used in offshore structural engineering (e.g. Huseby et al., 2013, 2015; Jonathan and Ewans, 2013). These contours are defined within the input variables space but are based on the probability of exceedance of the response variable. These methods rest on an approximation of the limit state curve and are independent from the model. The approach followed here was developed by Huseby et al. (2013, 2015). An environmental contour defined in this way is an (x,y) contour in the space (SWL, Hs), outlining a convex inner surface. The probability for the space outlined by the tangent to the contour and not containing the convex surface is constant (and equal to the probability for T) at every point along the contour (see Fig. 7):

$$P\big((H_s, SWL) \in \mathcal{B}^+\big) = \frac{1}{\lambda T} \tag{5}$$

We then need to find the maximum response Z along the contour, and this step is done identically to *jec* approach. The maximum response is then associated with a return period T, and written as $z_T^{enc}$.

Here, as in Bulteau et al., in prep, in considering these two methods to define scenarios it is assumed that in normal conditions, the two approaches (*jec* and *enc*) will calculate upper and lower boundaries of the true response $z_T$ and thus delineate the risk resulting from the propagation of forcing conditions from the open sea to the coast:

$$z_T^{jec} \leq z_T \leq z_T^{enc} \tag{6}$$





Figure 7

### 3.3.3 Covariates

Once the (Hs,SWL) combinations are identified for *enc* or *jec*, each Hs must be associated with a value for peak period and

peak direction.

In this study, only waves from the 60°-210° sector were retained (cf. above).

The normalised frequency of peak directions observed per Hs segment in the time series of peak Hs from the Candhis buoy (i.e. the sample of systematic data that was used to adjust the GPD law to the Hs with the HIVEBA method) shows that as from Hs > 2.75 m, the most probable peak direction is between 100° and 110° (Figure 1). The value Dp = 105° was therefore

retained for future simulations.

To model the peak wave period, we used an approach identical to that of Gouldby et al., 2014: the data for the peak period are first transformed into wave steepness by means of the equation:

$$St = \frac{2\pi H_S}{g T_p^2} \qquad (7)$$

Next, a conditional regression model in Hs, taking into account the heteroscedastic relationship between Hs and St whereby

the wave steepness tends towards a constant as Hs increases, is adjusted to the data (wave characteristics from the sample used to apply the H&T04 method). In the Monte Carlo simulation, a value for the wave steepness (and therefore the peak period) was thus associated with each simulated value for Hs (Figure 6b).

Based on the data from the Monte Carlo simulation and given that the pattern of change in the Tp *vs*. Hs relationship tends towards a deterministic law, it was decided to attach a single Tp value to each Hs produced by the combinations selected

from *enc* and *jec*, taking the median of the periods simulated for each significant wave height considered (Figure 6b).

### 3.3 Selecting multivariate scenarios

Table 5 shows the characteristics of the quadruplets (Hs,Tp,Dp,SWL) selected for *jec* and *enc* scenario respectively (see also Fig. 7). These scenarios were propagated via the digital modelling chain to estimate the response in term of the flood, represented by the extended of flooded area, volume of water in the inland or the maximum floodwater height.

Table 5

For each scenario, an instationary simulation was run for water level, overtopping and propagation of inland flooding. Based on an analysis of available water level and wave data, a 24-hour period was taken to simulate the storm conditions (including a 2h spin-up period for water level and wave conditions). The simulation time corresponds to the





duration of peak storm conditions regularly observed at the study site. For each scenario, the mean water level and wave dynamics at the Rank 0 limits are modelled following the shape of the 2013 storm, with concomitant water level and wave peaks at t+12h.

To analyse how flood risks would evolve with the mean sea level rise anticipated as a result of climate change, the scenarios were run with a uniform mean Sea Level Rise (SLR), with SLR=0 corresponding to current mean sea level conditions, SLR0.2 = SLR+0.2 m and SLR0.6 = SLR+0.6 m. The 0.2 m value of sea level rise was chosen in order to estimate the impact of a slight sea level rise (corresponding to a median scenario for 2046-2065 compared to the 1986-2005 global average (source: IPCC WG1 Ch13 - Church et al., 2013). The 0.6 m value corresponds to the mean sea level rise in the Mediterranean forecast by Slangen et al. (2014) for 2100. It should be stressed here that these are values chosen solely in order to demonstrate changing patterns of risk in scenarios for a gradual sea level rise, and that considerable uncertainties remain over the values for sea level rise in the Mediterranean, particularly because of the complex ocean processes taking place in the Gibraltar Straits.

## 4 Results

### 4.1 Simulating past events

Reproducing two different flood events makes it possible to assess modelling performance for water levels, overtopping volumes and the reproduction of water flows in the zones most affected during the events.

### 4.1.1 Simulating flood water levels

The water levels obtained by the simulations were compared with those deduced from the analysis of topographic landmarks photographed during the storms (on jetties, roads, etc.), whether affected or not (Figure 4). From different landmarks across the entire harbour zone, we were able to determine the mean water level in the harbour during the peak of the storm at 0.85 m IGN+/- 5 cm IGN in 2013 and at 1.05 m IGN+/- 5 cm in 2014 (Table 4).

The water heights in the harbour obtained by simulation are of a similar order to the water levels estimated from photographs (with a difference of less than 5 cm for 2013 and an under-estimation of about 10 cm for 2014). The wind action (maximum in 2013: 102 km/h, direction 90°N; in 2014: 89 km/h, direction 115°N) on the water height was slight, raising the water level by less than 5 cm in the harbour and 3 cm on the seafront during both events. However, the contribution of wave setup (maximum 27 cm in 2013 and 9 cm in 2014) appears to be a determining factor in reproducing water levels observed in the harbour. On the beaches, wave setup contributed up to 50 cm. Although we do not have the measurements needed to assess the quality of reproduction of wave setup and runup on the beaches. Nevertheless photographs taken during the storms show





that wave runup regularly overtop the berm on Port Leucate beach causing accumulation of water in back beach lows but do not produce overtopping at the sea front. Results of the SWASH model simulations concur with these observations. They show that with the given water level and wave conditions, wave runup overtop the first row of discontinuous dunes and fullfill back beach lows zone but without reaching the seafront. The qualitative relevance of the reproduction of wave runup

and overtopping during the 2013 storm is also supported by the results obtained for Zone B. In this sector, the first row of buildings sits directly on the upper beach, so that the seafront is affected by wave action during storms. The simulations produced results that concur with the observations of large overtopping volumes along the seafront.

### 4.1.2 Simulating breach flooding

The breach in the seawall during the 2013 storm was caused by both wave action and pressure due to the accumulation of

swash water on the structure.

We were not able to reproduce the consequences of the breach with the first method used, because the static water level reached at the height of the storm was below the level of the terrain where the breach occurred. The second method produced $V_{flood}$ levels that were quite close to the $V_{flood}$ levels deduced from compiled informations and GIS treatment.

The propagation of the instationary water volumes obtained with Method 2 shows the water height varying from 10 to 40 cm

and locally exceeding 50 cm (Figure 8a). The results of the simulation show that the extent of the flooded zones is consistent with observations, although slightly less extensive towards the south. These results also show an underestimation of water heights in the same zone. This is because regular overtopping by sheets of seawater in this sector is not taken into account. Simulations of overtopping only in this sector show considerable amounts of water entering the southern part of the neighbourhood (Figure 8b). When the propagation of water volumes flooding through the breach are coupled to simulation

of overtopping volumes across the seafront, the water heights in the southern zone are better reproduced (Figure 8c, Table 6). Water volumes from the breach in the seawall and from overtopping propagate trough the urbanized area controlled mostly by the topography. Consistent with observations, the combined water volumes flows towards the low-lying parts of the urbanised zone and then towards the natural area to the south, which is lower still. In the light of the information available, it appears that the method used overestimates the extent of the flooded areas (southern part of the neighbourhood). Although

this zone is known to have been flooded during the event but being a non-urbanized area no information exists against which the degree of overestimation can be assessed.

Figure 8

Table 6





## 4.2 Simulating 100-year return events

All of the joint 100-year scenarios (combined sea level/wave characteristics) were simulated in order to determine the scenario with the greatest impact for each of the two statistical methods used. The results were analysed in terms ($S_{flood}$), associated water volumes inland ($V_{flood}$) and water height ($H_{flood}$) in each of the three zones of the municipality (cf. Fig. 2).
The scenario with the greatest impact in terms of $S_{flood}$ and/or $V_{flood}$ is the Environmental Contour 1 scenario (ENC1). The scenario with the greatest impact using the *jec* method is JEC2 (Table 7).

Table 7

For both types of scenario (*jec* and *enc*), although the processes causing flooding in each zone are different in nature (overflowing and/or overtopping) and pattern (timing and coastal flooding patterns), the maximising scenario in both $S_{flood}$ and $V_{flood}$ response is the same (ENC1) for the 3 zones affected (A,B,C).
In the northern zone (Zone A), ENC1 triggers major flooding (in terms of flooded area) across the entire village with water depths generally below 0.50 m but exceeding 1 m locally (Fig. 9). Flooding in this sector is caused exclusively by overtopping across the seafront and affecting the area several hours at a stretch. The water then floods the entire neighbourhood, which is more low-lying than the seafront itself. The floodwater circulates through the whole southern part of the neighbourhood and overflows into the natural area to the south, filling the hollows.

Figure 9

With the ENC1-SLR0.2 and ENC1-SLR0.6 scenarios, the flooded areas (Sflood) reach further inland, especially towards the north. With the ENC1-SLR0.6 scenario, sheets of water sweep across almost the entire seafront of the neighbourhood, and overflowing occurs at the southern extremity. A change can be observed here in the nature of the processes causing flooding, which in turn significantly increases both water volumes (Vflood) and heights (Hflood). The Hflood values also almost systematically exceed 1 m.

In Zone B, flooding in the ENC1 scenario is also mainly associated with overtopping. As shown by historical observations, wave action rather than water height is liable to cause the most damage to the seafront. The zone behind it is flooded by the accumulation of overtopping water volumes. The inner edges of the neighbourhood, along the first line of buildings, are also affected by overflow flooding. The ENC1-SLR0.2 scenario shows that overtopping volumes are much larger along the seafront and also affect the southern part of the urbanized area. With the ENC1-SLR0.6 scenario, the situation is especially critical because, except for the southernmost part of the urbanised area, all buildings are affected by floodwater and access roads are submerged.

In Zone C, $S_{flood}$ values produced by the ENC1 scenario are fairly close to those observed during events in the recent past. Only the harbour zone is affected on the quays by $H_{flood}$ values of around 20-30 cm. Given the width and morphology of the





beach (cf. Fig. 2, profiles 8 to 11), the overtopping sheets of seawater do not reach the seafront buildings (or only slightly in the southern part). With ENC1-SLR0.2, the 0.2 m rise in sea level increases the number of sectors submerged by overflowing floodwater and also accentuates potential overtopping along the seafront.

With a sea level rise of 0.6 m (ENC1-SLR0.6), $S_{flood}$ extends over a much larger area in zones along the harbour (overflow

flooding), with several sectors under more than 0.5 m of water. Floodwater from overtopping propagates from the seafront to the lower parts of the lido, then into numerous areas in the north and centre of Zone C. In the southern part, the overtopping water volumes accumulate in the natural area, submerging roads under several decimetres of water (cf. fig 9).

The simulations with a mean rise in sea level show the extent to which the site is affected in general by threshold effects: with ENC1-SLR0.2, $V_{flood}$ increases by 188% and $S_{flood}$ by 160%. With ENC1-SLR0.6, the situation is critical, with a 384%

increase in $V_{flood}$ and 247% in $S_{flood}$.

## 4.3 Overflowing vs overtopping flooded area

Identifying the zones affected by each the two types of flooding shows why both types, overflow ($O_{flow}$) and overtopping

($O_{topp}$) have to be taken into account to show and characterize the exposure of the Leucate municipality to flood risks (Fig. 10, Fig. 11).

Figure 10

Zone A is affected exclusively by $O_{topp}$ with the scenarios where mean sea level rise is less than 20 cm. When only the

maximising scenarios for each statistical method (ENC1 and JEC2) are considered, overflow flooding occurs only in scenarios with an mean sea level rise of +0.60 m and represents only 11% and 15% of inland $V_{flood}$.

Flooding in Zone B is also mainly by overtopping along the seafront, affecting 77% (JEC2) to 84% (ENC1-SLR0.6 and JEC2-SLR0.6) (cf. Table 7). The $O_{topp}/O_{flow}$ ratio is fairly stable considering the different SLR scenarios.

In Zone C, the $O_{topp}/O_{flow}$ ratio changes considerably with the different scenarios. With the ENC1 and JEC2 scenario, for

example, almost all $V_{flood}$ (97 and 98%) is caused by overflow. The ratio between $V_{flood}$ triggered by $O_{flow}$ and $O_{topp}$ is significantly different in the scenarios with mean sea level rise of +0.20 and +0.60 m: $O_{flow}$ is still the main process associated to flooding although $O_{topp}$ accounts for about 1/3 of $V_{flood}$ with the SLR0.2 scenarios and a little under 1/4 of $V_{flood}$ with the SLR0.6 scenario. These differences show on the one hand that the characteristics of flooding process are significantly changing with SLR scenario on the second hand respective contributions do not change linearly and that they

depend on topographic particularities and threshold effects.

Figure 11

The simulations run for scenarios with no mean sea level rise show that at present, the majority of coastal flooding in the municipality is due to overtopping ($O_{topp}$). The low-lying areas affected directly by $O_{topp}$ and indirectly by the propagation of





the resulting water volumes account for 62% of $S_{flood}$ (38% of these sectors are flooded by $O_{flow}$). A moderate sea level rise of less than +0.2 m does not affect this distribution of flooding patterns ($O_{topp}$ = 63% as against $O_{flow}$ = 37%). However, a larger rise in mean sea level of +0.60 m (by 2060-2080 in the IPCC's BAU scenario) significantly affects the ratio between sectors flooded by $O_{topp}$ and $O_{flow}$, which for the municipality as a whole tends to equalise, with a ratio for $S_{flood}$ of $O_{topp}$ =

54% and $O_{flow}$ = 46%.

### 3.4 Determining the 100-year uncertain flood area

The two statistical methods selected to build up the scenarios, i.e. different combinations of offshore marine forcing conditions with a given return period, can - once propagation has taken place - produce significantly different results.

In this section, we will therefore analyse the differences in $S_{flood}$ and $H_{flood}$ obtained after simulating the scenario with the greatest impact defined with each of the statistical methods used and on the assumption of a mean sea level rise of +0.6 m (ENC1-SLR0.6 and JEC2-SLR0.6 scenarios) (Fig. 12). The illustration proposed here focuses on the central part of Zone C, because in the built-up sectors in the other zones, the differences in extent and water height are relatively slight (mostly less than 0.1 m with both scenarios considered, ENC1 and JEC2). Indeed, most of the differences across the municipality are of

less than 0.1 m, which may be considered as not very significant. This order of uncertainty is identical or below that obtained when comparing levels produced by modelling and actually observed during recent events. Furthermore, LIDAR topographic data are usually characterised by errors below 0.2 m. We have therefore considered that the uncertainty associated with the statistical method chosen is not significant for the zones shown in blue (Fig. 12).

20                                              Figure 12

However, as Fig. 12a shows, both $S_{flood}$ and $H_{flood}$ can differ significantly in Zone C depending on the scenario. For example, differences in $S_{flood}$ can be observed that are related to the statistical method used (zones in red in Figure 12a). Here, the zones in red are considered to be zones of "uncertainty" as regards characterisation of the hazard. These sectors are not

greatly flood-prone, if at all, with a JEC2 scenario but may be subject to $H_{flood}$ of 0.1 m to more than 0.5 m with ENC1.
These differences may be considered as moderate (green and yellow from 0.1 m à 0.3 m) to large (red zone from 0.3 m to 0.5 m), and show that the risk intensifies considerably in the zones subject to threshold effects (topographic hollows). Significant differences between the JEC2 and ENC1 scenarios were observed in Zone B, and considerable differences in Zone C with the JEC2-SLR0.6 and ENC1-SLR0.6 scenarios.

Looking now at the marine forcing used for the two types of scenarios, a difference of 0.04 m in the offshore sea level and of 0.4 m and 0.3 s respectively for Hs and Tp (i.e. a difference of about 5% in the forcing conditions) produces differences in





$H_{flood} > 0.3$ m in some streets in the town centre subject to $O_{flow}$ and $O_{topp}$ risks. In other words, the response in terms of flooding is highly sensitive to variations in the parameters chosen, especially when a rise in mean sea level is considered. The differences for total $V_{flood}$ and $S_{flood}$ show that a variation of about 5% in the forcing parameters results in $V_{flood} = +13.5\%$ and $S_{flood} = +11.3\%$. With the SLR0.6 scenario, the relative differences (with $V_{flood} = +8.5\%$ and $S_{flood} = +5.3\%$) become

smaller because all of zones A, B and C are flooded.

Without making an analysis of the sensitivity of the linked models to forcing parameters, which was not the object of this study, our interpretation is as follows: given the statistical approaches used to determine the forcing scenarios to be propagated, one considered to be minimising (jec) and the other maximising (enc), we can consider that if $S_{flood\_jec} = S_{flood\_enc}$, the zone is very likely to be subject to a 100-year flood risk (zones in blue, Fig. 12a). Given the generally small differences

in these zones, with $H_{flood\_jec} = H_{flood\_enc}$ (±0.1 m), we can also consider that the assessment of water heights is satisfactory. However, the zone in red can be considered as a zone of uncertainty in defining the 100-year risk.

Considering the hazard characterisation for the all study area, the $H_{flood}$ uncertainty arising from the statistical method used translates into a moderate impact on the spatial extent of flooding. However, the differences locally can be considerable, radically changing the nature of the risk.

These differences are due above all to threshold effects, when a small change in water height exceeds a topographic threshold and allows propagating a great deal of water inland, which accumulate in topographic hollows. In our case study, these zones are mainly located in Zones B and C. In the latter, they only become very evident with the SLR0.6 scenarios.

## 5 Discussion

The work undertaken to characterise the flood hazard at the Leucate site is the outcome of a succession of approaches. The first was to apply the recommendations of the French Risk Prevention Plan (PPR) using a fixed elevation and available observations from tide gauges along the French Mediterranean coast (DREAL LR, 2008). Subsequently, Anselme et al. (2011) showed that the additional water height caused by wave setup and runup has to be taken into account to approach the values observed during past storms and to characterise the risks to the seafront. However, the parametric method applied

cannot be used to consider the risk in zones not directly exposed to waves, such as harbour zones where the flooding pattern is different ($O_{flow}$). Our study shows that to map flood risks, it is just as important to consider the overflow ($O_{flow}$) risk as potential overtopping water volumes ($O_{topp}$).

The method applied in this study allowed the $O_{flow}$ hazard to be addressed by adding the wave setup contribution into the mean water level reach during the storm. The contribution at the storm surge due to wave setup can reach 50 cm on the

beaches and 25-30 cm in the harbour, making it a decisive factor to address flooding along the inner part of the lido (up to 1/3 of the total rise).



The simulations to reproduce two events in the recent past produced a satisfactory representation of water levels in the harbour (average underestimation of 5 cm for 2013 and 10 cm for 2014). Besides the errors inherent to the simulation method, there may be several reasons for the differences of a few centimetres that appeared between observations and modelling results. These include a lack of precise forcing data, the used of fixed bathymetry and potential resonance effects

in the harbour that are not reproduced by the models used. Furthermore, sea levels in the northernmost pass are substantially underestimated (by 0.25 to 0.30 m). The underestimation is mainly due to the narrowness of the pass (15 to 20 m) and the potentially highly changeable bathymetry. These characteristics are the reason for the poor reproduction of water flows and levels in this sector, but do not appear to alter the results for the other sectors in the studied area.

On the other hand, the chain of models was able to handle zones potentially affected by overtopping by estimating the water

volumes liable to overtop the seafront. As in the studies in the bibliography, the information from recent events against which the reproduction of overtopping volumes was assessed for accuracy is less detailed for natural zones (few observers) and not easily quantified (overtopping simultaneous with overflow or taken together with rainwater flows). The simulations run did however indicate where overtopping occurred (to the south of Leucate Plage, north of the naturist village) or did not occur (Port Leucate beach), concurring with the available qualitative information (wave damage to the seafront, eyewitness

accounts). However, this information is not sufficient to assess whether the reproduction of the overtopping volumes is accurate. This highlights the need to produce accurate validation data (cf. Gallien et al., 2016) to assess $O_{topp}$ on the field. It would be necessary, during future storms, to establish measuring protocols based on video data and topo-bathymetric monitoring data before and after the storm, in order to collect more precise data that would help to identify sectors subject to $O_{topp}$.

The extreme values analyses undertaken in this study to define scenarios for propagation are innovative in two respects. First, by using a Bayesian approach (HIBEVA method), we were able to combine data of different types and different levels of accuracy, and thus to calculate the marginal probability distribution for Hs and consider long return periods. This would not have been possible by using only Candhis observation data, as the uncertainties over the estimated values would have been too great for return periods of more than 30 years. Secondly, the definition of offshore forcing scenarios to estimate

100-year coastal flooding hazards was based on two different statistical methods, one producing joint exceedance contours and the other environmental contours. The advantage of using the two methods is that while it is not possible to make a precise assessment of the 100-year flood risk (there is not enough data on flooding available to analyse the extreme values of response variables directly), it is possible to bracket the 100-year flood risk between the values for the response variables ($V_{flood}$, $S_{flood}$ and $H_{flood}$) that result from propagating the scenarios chosen with the two methods (cf. Equation 8). This also

gives an indication of the robustness of the result. For example, in our case study, the built-up areas in Zones A and B are not very sensitive to the statistical method chosen, which indicates a sufficiently high level of confidence in the estimation of the 100-year hazard in these zones. For Zone C, on the other hand, there are notable differences depending on the statistical method applied, reflecting a greater uncertainty in the estimation of the 100-year hazard for several neighbourhoods. To overcome this uncertainty arising from the choice of scenarios for propagation, one possible solution is to use a meta-model



which is, in essence, a mathematical approximation of a hydrodynamic model that predicts the modelled responses at a negligible cost in computing time (Idier et al., 2013). In this way, it becomes possible to estimate the response variables directly by "propagating" all the simulated combinations of forcing conditions obtained from the Monte Carlo simulation (cf. §2.4.1). This type of approach has been used in the coastal engineering field for regular and continuous modelling (Camus et

al., 2011; Idier et al., 2013; Gouldby et al., 2014; Rueda et al., 2015). Unfortunately, in our case study, the complexity of the modelling chain prevents the use of classic meta-modelling techniques, and developing new alternatives is beyond the scope of this study.

The differences in $H_{flood}$ between scenarios JEC2 and ENC1 show that threshold effects are liable to notably change the nature of the hazard, with sectors where small differences in forcing (around 5 %) can cause differences in water levels of 30

to 50 cm. It should be remembered here that modelling the inland propagation of coastal flooding is based on significant efforts to integrate terrain roughness, buildings, obstacles and flows and, conversely, on controlling the continuity of flows along the main traffic routes. However, as the propagation models are set at a spatial resolution of 5 m, they may trigger a threshold effect in some sectors (narrow street, topographic irregularity, etc.).

## 15  6 Conclusion

Using a modelling method based on a chain of several MARS-SWAN-SWASH models, we were able to reproduce water levels, $O_{flow}$ and $O_{topp}$ for two recent events consistently with the quantitative and qualitative information available for the site.

Scenarios for the forcing conditions of joint 100-year return periods were determined by means of two different statistical

methods (joint exceedance contours and environmental contours) in order to analyse the differences arising from the method used to define the scenarios. Simulations of the different 100-year scenarios show that the choice of statistical method used to define the forcing conditions for the scenarios produces notable differences in the response variables considered ($V_{flood}$, $S_{flood}$ and $H_{flood}$). The largest differences are in Zone B with a sea level scenario based on the current mean sea level, and in Zone C with a mean sea level rise of +0.6m.

Because the *jec* method is minimising and *enc* maximising, using the two types of scenarios enabled us to calculate minimum and maximum values for the spatial extent and height of floodwater, thus bracketing the 100-year hazard. This also enabled us to characterise the uncertainty over the results that arises from the type of scenario chosen: whereas the results are robust when the $S_{flood\_jec}$ response = $S_{flood\_enc}$ and $H_{flood\_jec}$ = $H_{flood\_enc}$ (± 0.1 m), the uncertainty is greater when these conditions are not met. In some sectors, this uncertainty can translate into differences of 0.3 to 0.5 m. The simulations

of the different scenarios also bring out two major characteristics of the flood risk in the Leucate municipality.



The first is that the types of flooding that affect the municipality are spatially different. This means that a realistic appraisal of the risk requires joint simulations of flooding by overflow and overtopping. With a maximising 100-year hazard scenario, for the municipality as a whole, 38% of the zones are prone to overflow flooding and 62% to flooding by propagation of overtopping water volume along the seafront.

The second is that the nature and scale of the hazard is likely to evolve drastically as the mean sea level rises. For a 100-year event, our results show that overflow flooding affecting built-up zones is limited in extent. The hazard mainly arises from overtopping along the seafront, which is likely to cause significant flooding in the northern part of the municipality (Zone A). Although the hazard increases with a scenario based on a +0.20 m mean sea level (SLR0.2), the newly affected zones are mainly natural areas or roads, with little change in the characteristics of the hazard (ratio between zones affected by overflow

flooding / overtopping). On the other hand, the SLR0.6 scenarios illustrate what is meant by a tipping point (Sweet and Park, 2014), since they produce a 250% increase in flooded areas in a 100-year hazard situation, with flooding across the entire municipality, built-up sectors severely affected by overflow flooding (Zones A and C) and traffic and evacuation roads becoming almost impassable.

A further point to be made here is that this study focused only on the consequences of climate change under different

assumptions of mean sea level rise. It did not address the consequences of potential changes in marine conditions (waves) or of an intensification of weather conditions during storms. Given the current exposure of the study site to wave overtopping, scenarios assuming an increase in storm intensity (atmospheric surge or wave conditions) would most certainly lead to more intense flooding by overtopping waves and exacerbate the flood hazard in general.

These changes in the flood hazard and especially in the ratio between zones subject to flooding by overflow and/or

overtopping will not only alter the structural vulnerability of urban areas but also require changes in the messages to be communicated to the public on flood risk awareness and steps to be taken for crisis management in case of flooding event.

**Acknowledgements.**

We would like to thank the Conseil Supérieur de la Formation et de la Recherche Stratégiques (CSFRS) for the funding of the CRISSIS research program that initiated this publication. We would also like to thank the municipality of Leucate, which

has agreed to be a partner in the program. We thanks particularly Nicolas Guilpain and Bruno Troqueraud from the capitainerie of Port Leucate, for the help in the field. We would also like to thank all members of the CRISSIS research program. Finally, thanks to Yves François Thomas who initiated this project and whose teachings allowed the original approach of the marine flood hazard developed in this work.

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





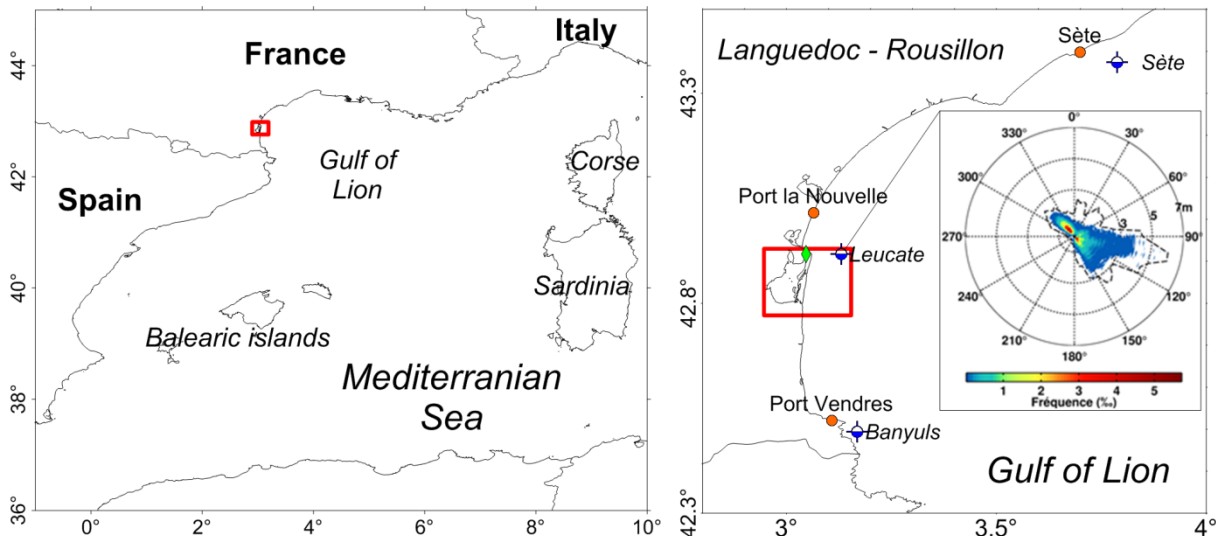

**Figure 1 : Location map, on the left, and zoom-in on the right. Circle for tide gauge location, cross for buoy location and diamond for Leucate meteorological station. The red rectangle delimits the domain used for simulation (R0).**

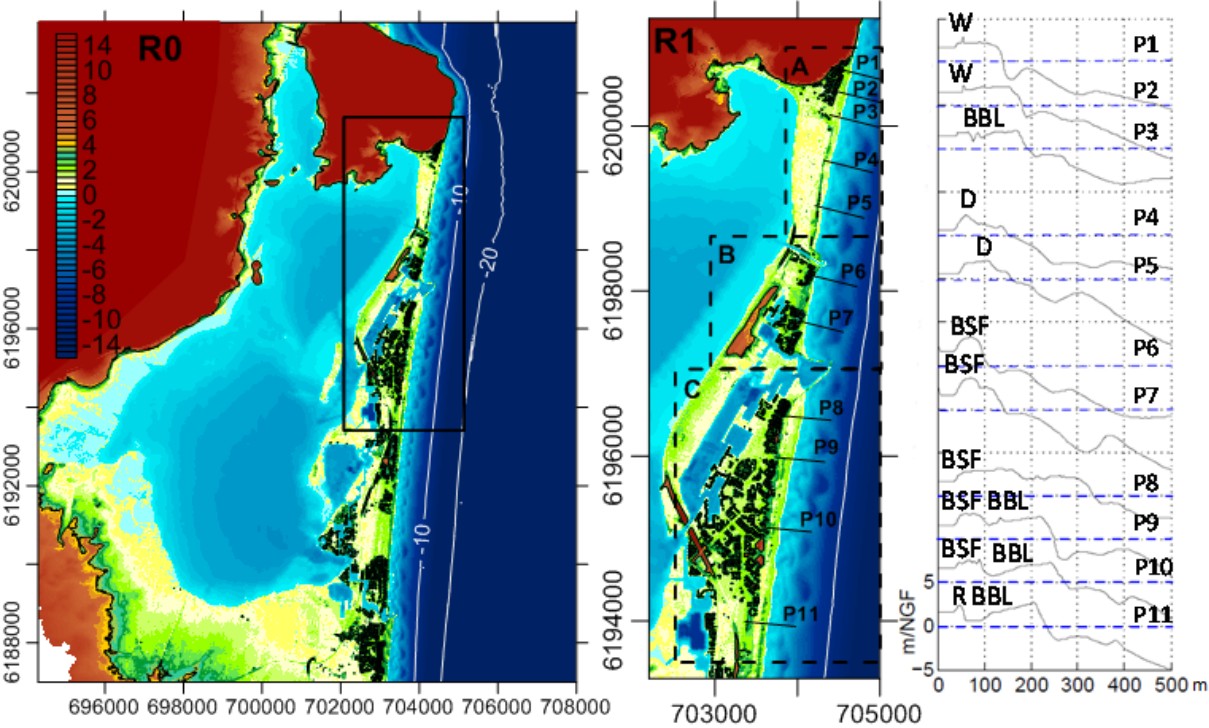

5 **Figure 2 : Study site and simulating domains : R0 (extension 16.5 x 16.5 km) resolution 20 m, R1 (extension 3 x 8 km) resolution 5 m and 11 topo-bathymteric profiles over three studied zones (A: Leucate Plage; B: the naturist village; C: Port Leucate). Mains sea front characteristics are presented as (W) for sea Wall, (BSF) for Built Sea Front, (D) for Dune, (BBL) for Back Beach Low, (R) for Road.**




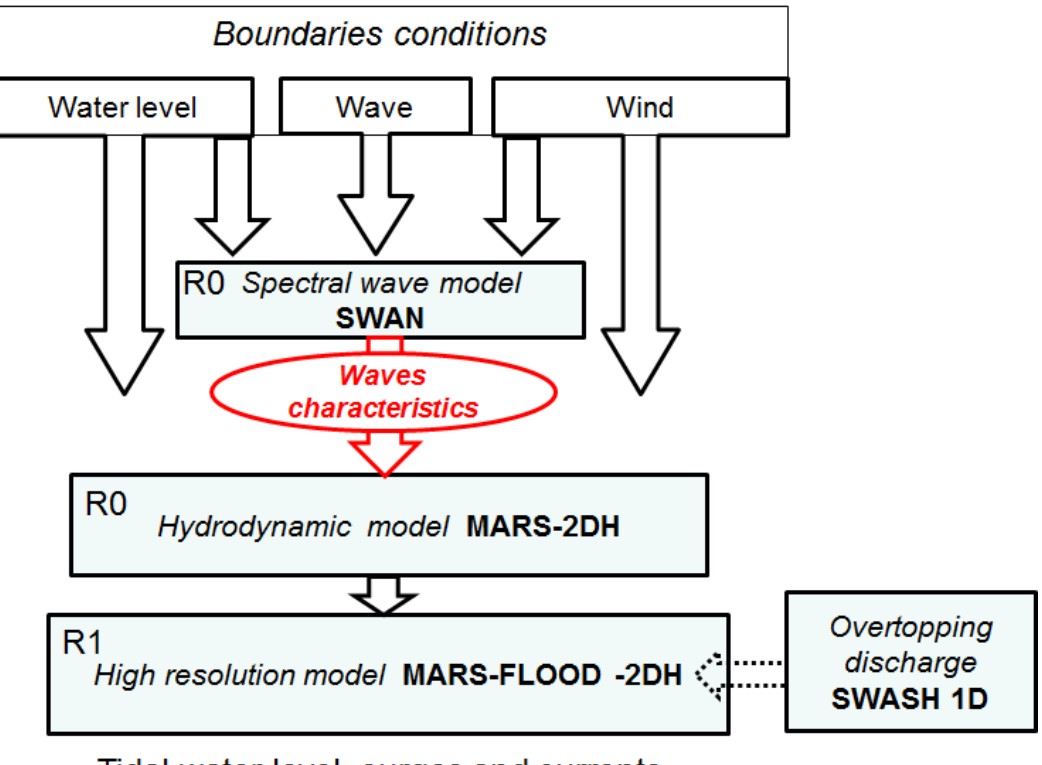

**Figure 3 : Chained modelling method**




**Figure 4 : Example of "hard" information relative to water level during the 2013 and 2014 storm events. Reached water levels on pictures were measured on field using D-GPS in order to estimate quantitative water level. Red point are related to 2013 event and Blue to 2014 even information. (photograph source : Leucate municipal agents)**




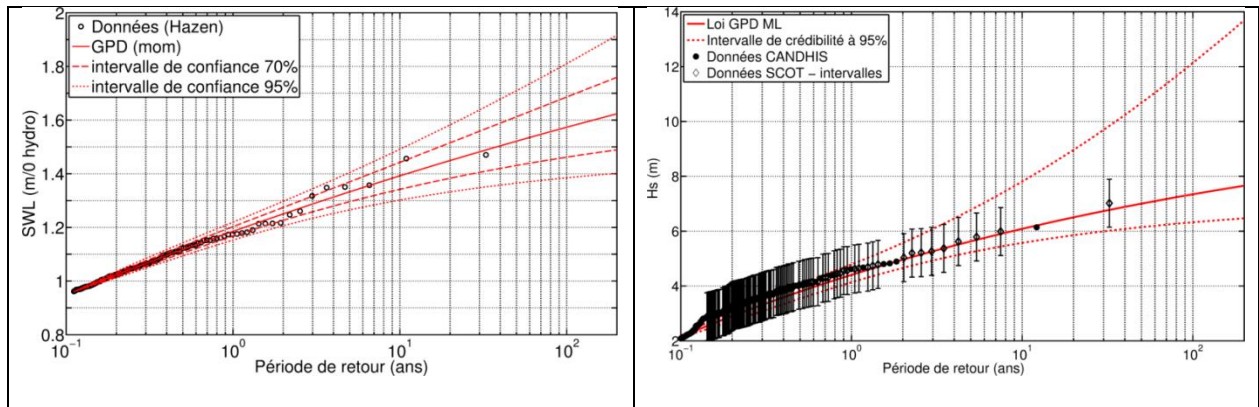

**Figure 5**: On the left, a) GPD adjusted to the Sète tide gauge data. Threshold of the law is fixed at 0.96m Z.H. Parameters of the law are estimated using the method of moments. Confidence intervals are calculated by parametric bootstrap (Mazas and Hamm, 2011). On the right, b) GPD law adjusted to Hs data (Candhis et SCOT) by the HIBEVA method. The threshold is fixed at 2m. For illustration, the SCOT data are presented by the central values of each interval.

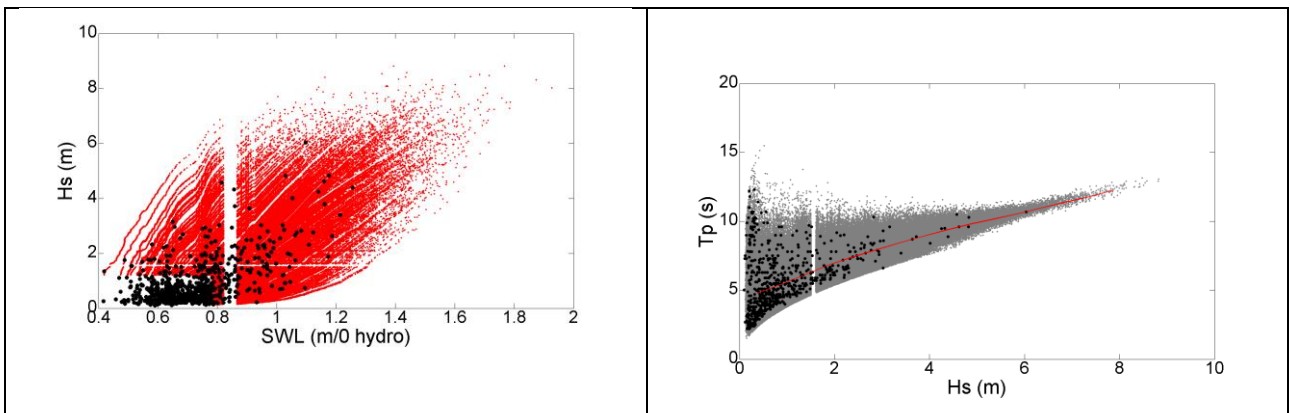

**Figure 6:**a) On the left, a) results of Monte Carlo simulation for variables Hs and SWL based on 6 common years between SWL data and Hs data (Candhis). Declustered data in black, simulated data (10000 years) in red. On the right, b) results of Monte Carlo simulation for variables Hs and Tp. Black dots: declustered data. Grey dots: simulated data. In red : median of the periods simulated given Hs.





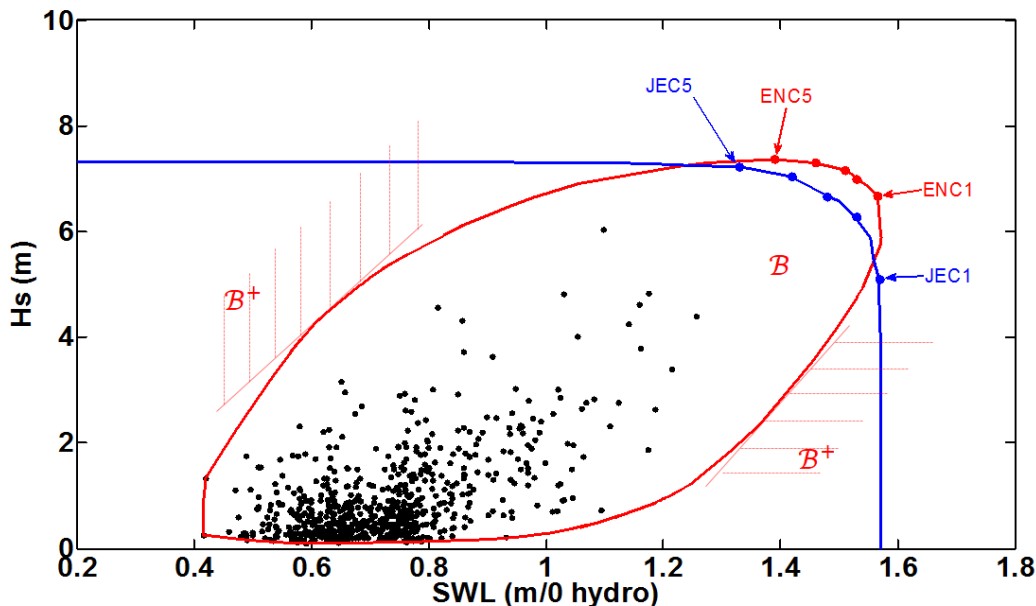

**Figure 7 : 100-year JEC (blue) and ENC (red). $\mathcal{B}^+$ is the surface delineated by the tangent to the contour and which does not contain the convex surface $\mathcal{B}$. The tangent is a linear approximation of the true limit state function (Huseby et al., 2013).**





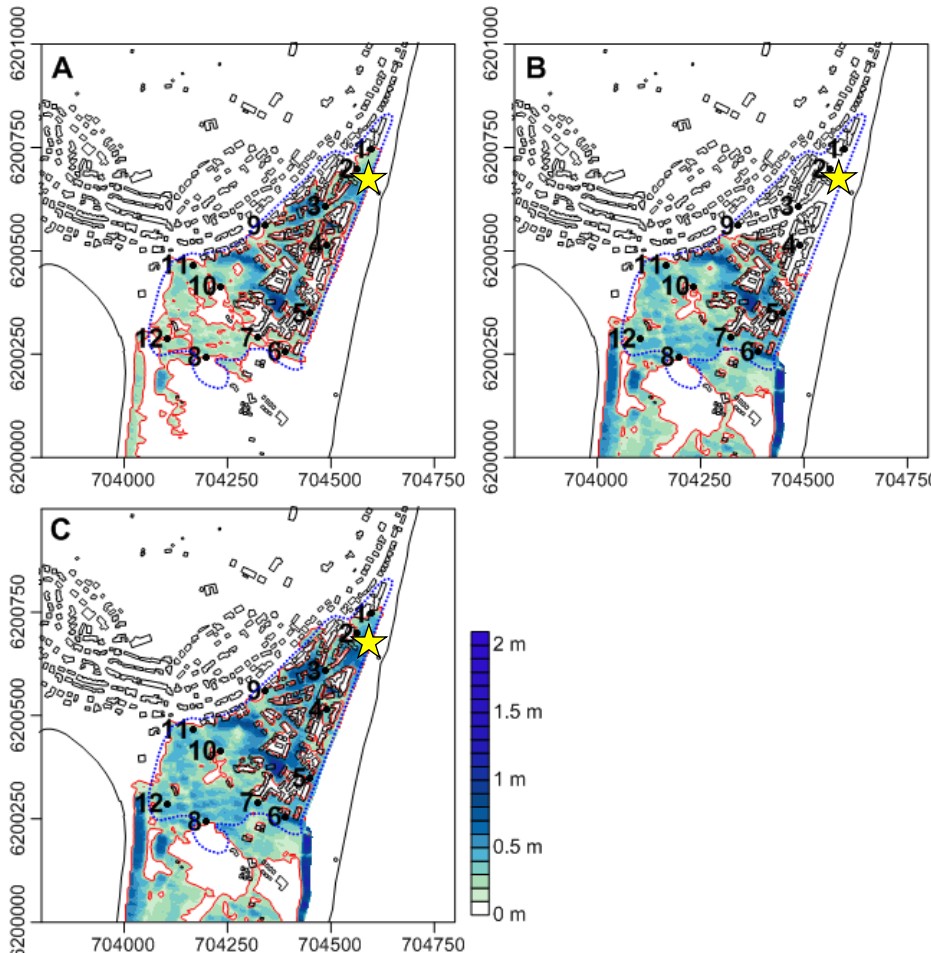

**Figure 8 : Flood simulation results for the zone 1 (Leucate Plage) after sea front wall breach (the yellow star indicate the location of the breach). The blue dotted line represents the reconstructed flood extension, the red line the simulated flood extension. A) Propagation of the water volume passing through the breach, B) propagation of overtopping water volume, C) Propagation of the two source of water.**





**Figure 9 : Results for the most impacting 100 years scenario, for ENC1, ENC1-SLR0.2, ENC1-SLR0.6.Red line represents the flood extension**




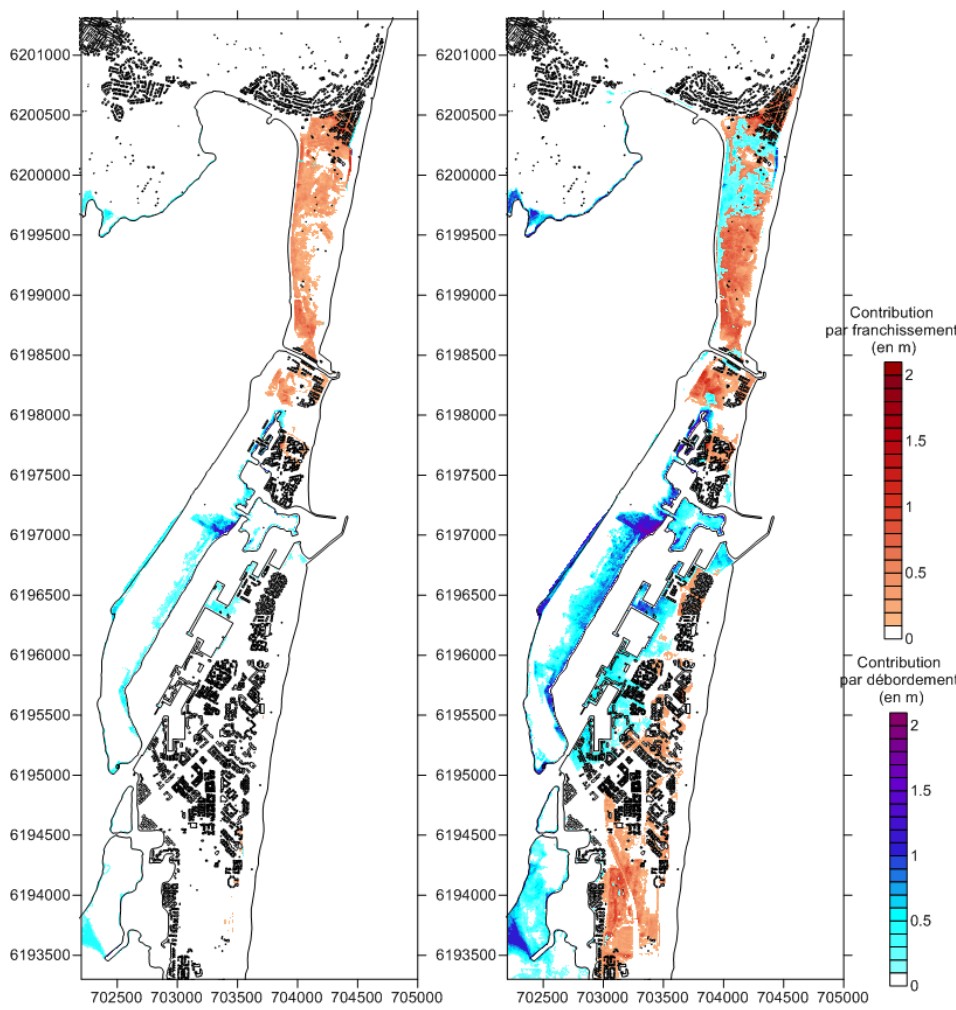

**Figure 10 : Affected area by overflowing and overtopping for ENC1 and ENC1-SLR0.6 scenarios**




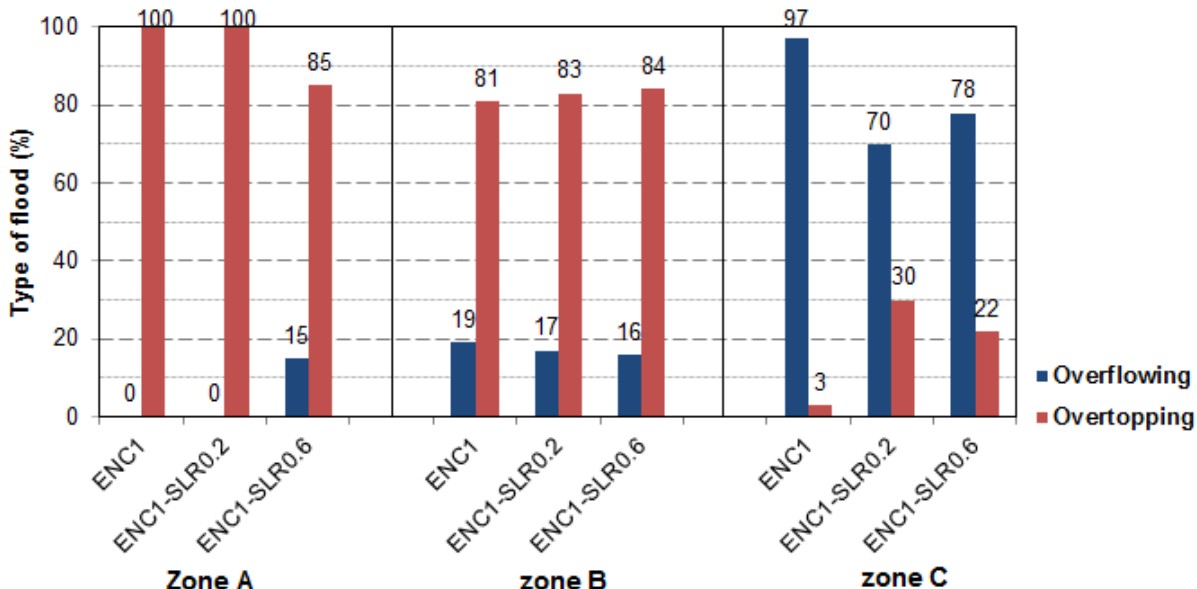

Figure 11: Respective contributions of overflowing and overtopping processes in total flood surface (%)

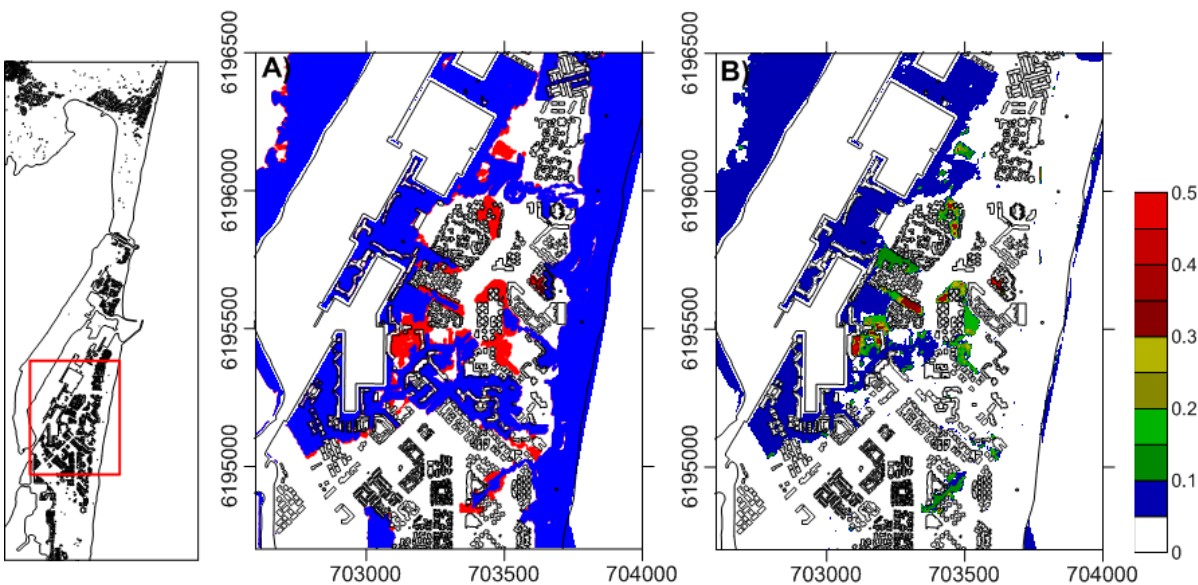

5    Figure 12 : Differences between ENC1-SLR0.6 and JEC2-SLR0.6 in the C zone. A) Maximal extension of flooded area, blue surface are flooded for twice scenarios, red surfaces are only flood for ENC1-SLR0.6 scenario. B) Differences in water level





**Table 1 : Observed and simulated data base**

| Data | | Historical events | | Statistical analysis |
|---|---|---|---|---|
| | | March-13 | November-14 | 100-years event |
| **Water level** | observed | Sète tide gauge | PLN* tide gauge | Sète tide gauge 1996-2015 |
| **Wave** | observed | Leucate Buoy | Leucate Buoy | Leucate Buoy 2007-2015 |
| | simulated | MEDNORD | MEDNORD | GuLWa 1979-2009 |
| **Wind** | observed | Leucate station | Leucate station | |

*PLN : Port la Nouvelle.

**Table 2 : Topo-bathymetric data base**

| Data | Location | Source | Data type | Spatial resolution/vertical precision |
|---|---|---|---|---|
| **Bathymetric** | Offshore (> 10 m depth) | SHOM | Probes | 20 m / decimeter |
| | Neashore (< 10 m depth) | Litto3D (SHOM-IGN) | LIDAR MNT | 1 m / centimeter |
| | Port and pass | DREAL LR 2001, 2003 Mesuris 2012 Asconit 2012 | Mono and multi-beam survey | 10 cm / centimeter |
| | Lagoon | IFREMER, 2001 | Mono and multi-beam survey | 10 cm / centimeter |
| **Terrestrial** | Coast | Litto3D (SHOM-IGN) | LIDAR MNT | 1 m / centimeter |
| | Building | BD Topo (IGN)/ Litto3D (SHOM-IGN) | LIDAR MNE | 1 m / centimeter |
| | Coastal structure | Field campaign | D-GPS | 1 cm / centimeter |

**Table 3 : Used Stickler coefficient**

| Land | Strickler coefficient |
|---|---|
| Pine forest | 10 |
| Forest | 10 |
| Dune with bushes | 15 |
| Agricultural area | 17-20 |
| Dune with vegetation | 25 |
| Sand | 33 |
| Urban green space | 33 |



| | |
|---|---|
| Industrial area | 40 |
| Sea floor | 40 |
| Asphalt | 67 |

**Table 4 : Observed vs simulated water level, qualitative and deduced quantitative information.**

| Storm | Location | Observations | Deducted water level (m/NGF) | Simulated water level (m/NGF) | Différence (m) |
|---|---|---|---|---|---|
| **2013** | Pt13_1 | Height of the quay | 0.85-0.90 | 0.94 | 0.05 - 0.1 |
| | Pt13_2 | No quay ovreflowing | 0.85- 0.90 | 0.82 | 0.05 - 0.1 |
| | Pt13_3 | Quay overflowing | 0. 85 | 0.81 | 0.05 |
| | Pt13_4 | No quay ovreflowing | 0.80– 0.85 | 0.58 | 0.2 – 0.25 |
| **2014** | Pt14_1 | Quay overflowing | 1.05– 1. 10 | 0.92 | 0.1 – 0.2 |
| | Pt14_2 | Quay overflowing | 1–1.05 | 0.92 | 0.05 – 0.15 |
| | Pt14_3 | Quay overflowing | 1– 1.05 | 0.92 | 0.05 – 0.15 |
| | Pt14_4 | Quay overflowing | 0.95–1.05 | 0.94 | 0.05 – 0.1 |

5    **Table 5 : Combinaisons from contour CDC centennal (CB 1 to 5) and contour CE centennal (CB 6 to 10) discretisation**

| | Combinaisons | | | | |
|---|---|---|---|---|---|
| | **JEC1** | **JEC2** | **JEC3** | **JEC4** | **JEC5** |
| **Hs (m)** | 5,09 | 6,27 | 6,66 | 7,04 | 7,22 |
| **Tp (s)** | 10,0 | 10,9 | 11,2 | 11,5 | 11,7 |
| **Dp (°)** | 105 | 105 | 105 | 105 | 105 |
| **SWL (m/NGF)** | 1,14 | 1,10 | 1,05 | 1,01 | 0.92 |
| | **CE1** | **CE2** | **CE3** | **CE4** | **CE5** |
| **Hs (m)** | 6,67 | 6,98 | 7,15 | 7,30 | 7,37 |
| **Tp (s)** | 11,2 | 11,5 | 11,6 | 11,7 | 11,8 |
| **Dp (°)** | 105 | 105 | 105 | 105 | 105 |
| **SWL (m/NGF)** | 1,14 | 1,10 | 1,08 | 1,03 | 0.96 |

**Table 6 : Observed vs simulated water level reproducing flood after breaching**

| | P1 | P2 | P3 | P4 | P5 | P6 | P7 | P8 | P9 | P10 | P11 | P12 | mean |
|---|---|---|---|---|---|---|---|---|---|---|---|---|---|
| **Observed water level** | 0.1 | 0.2 | 0.3 | 0.3 | 0.3 | 0.2 | 0.3 | 0.1 | 0.1 | 0.4 | 0.4 | 0.3 | |
| **Breach with SWASH** | 0.11 | 0.21 | 0.44 | 0.24 | 0.33 | 0.2 | 0.13 | 0.05 | 0.1 | 0.34 | 0.27 | 0.13 | |





| | | | | | | | | | | | | | |
|---|---|---|---|---|---|---|---|---|---|---|---|---|---|
| **Difference (m)** | 0.01 | 0.01 | 0.14 | -0.06 | 0.03 | | -0.17 | -0.05 | 0 | -0.06 | -0.13 | -0.17 | -0.04 |
| **Breach SWASH + overtopping** | 0.11 | 0.21 | 0.44 | 0.24 | 0.45 | 0.6 | 0.36 | 0.07 | 0.13 | 0.39 | 0.39 | 0.3 | |
| **Difference (m)** | 0.01 | 0.01 | 0.14 | -0.06 | 0.15 | | 0.06 | -0.03 | 0.03 | -0.01 | -0.01 | 0 | 0.03 |

**Table 7 : Flooded surface and water volume for all the combinations relatively to the maximizing combination (CB 6) in %**

| | JEC1 | | JEC2 | | JEC3 | | JEC4 | | JEC5 | |
|---|---|---|---|---|---|---|---|---|---|---|
| | Surfaces | Volumes | **Surfaces** | **Volumes** | Surfaces | Volumes | Surfaces | Volumes | Surfaces | Volumes |
| **Zone A** | 84 | 75 | **96** | **92** | 94 | 92 | 85 | 79 | 61 | 48 |
| **Zone B** | 40 | 35 | **82** | **75** | 79 | 72 | 69 | 58 | 42 | 36 |
| **Zone C** | 73 | 75 | **81** | **82** | 76 | 76 | 66 | 67 | 53 | 53 |
| | **CE6** | | CE7 | | CE8 | | CE9 | | CE10 | |
| | **Surfaces** | **Volumes** | Surfaces | Volumes | Surfaces | Volumes | Surfaces | Volumes | Surfaces | Volumes |
| **Zone A** | **100** | **100** | 96 | 94 | 94 | 89 | 88 | 78 | 62 | 50 |
| **Zone B** | **100** | **100** | 97 | 95 | 89 | 87 | 77 | 69 | 35 | 27 |
| **Zone C** | **100** | **100** | 96 | 95 | 92 | 92 | 80 | 80 | 63 | 65 |