# Peer review of "High-resolution marine flood modelling with coupled overflow and overtopping processes: framing the hazard based on historical and statistical approaches."

_Natural Hazards and Earth System Sciences, 2017_

## Referee Comment (RC1) · Anonymous Referee #1 · 5 Jun 2017

1. Line 11, expected frequency of occurrence or the probability that the event will be exceeded in any year.

2. Line 25, Expand the term SWASH model.

3. Line # 15 – 21, authors pointed that it does not require any assumption of dependence structure and extended to higher dimension. However, they only multivariate GPD model in their analysis. On the other hand, the marginal variables can follow any distributions, and copula modeling offers this advantage over other multivariate distri-

butions. In addition, they also offers tail dependency through various metrics, such as, CFG, LOG or SS estimators. Extension to higher dimension is feasible using Vine or t-copulas.

4. Line 19, is the peak period unit is in 'seconds'?

5. Some of the references are missing, such as, Stepanian et al. 2014, Lazure and Dumas, 2007. Please check others.

6. Line 25 ∼ 30, will land cover and land use data remain stationary since 2006 even if considering impact of urbanization?

7. Line 6, 'Nord' is to be replaced with 'north'.

8. Line 7, 'IFREMER MEDNORD' model. In summary, it would be good to give short description of the various models used in this analysis either in the Appendix or in Supplements. Further, in all cases, only abbreviation of the model names are used. It would be nice to include full model name for the first time and use the abbreviation subsequently.

9. Line 31, please provide information regarding temporal resolution of the data here. The period of data (1996-2015) used are of 20-years, but it is mentioned as 16.4 years. Is this due to a few years of missing data?

10. Line 4 – 5, no autocorrelation, cross correlation analyses are shown to prove independence.

11. Wave angles between 60° and 210° were kept for the analysis. Do they represent extreme scenario?

12. Does threshold of the GPD model is kept as a fixed value throughout or it is considered as variables based on moving window time frame?

13. Line 14, "To finish" could be replace with "Finally" or "The next"

14. The joint exceedance can be calculated using 'AND' and 'OR' cases. In general, the joint probability is underestimated in 'OR' case and overestimated in 'AND' case. When author(s) said 'the response variable is an underestimation' I presume calculation of return periods are performed using 'OR'-case. Refer foll. for details: Ganguli and Reddy (2013), Probabilistic assessment of flood risk using trivariate copulas. Theoretical and Applied Climatol.

15. Line 26, explain the term 'instationary' in this case. In general, instationary refers to transient or quasistationary. Do authors perform nonstationary simulation in this case?

16. To analyze impact of climate change, a scenario-based analysis is performed based on earlier literature, which might carried out using older generation climate models. It would be nice to see climate change impact using finer resolution regional climate models considering future change using set of RCP scenarios after employing an appropriate bias correction scheme.

17. Page 16, Line 2, "Results of the SWASH model simulations concur with these observations". Refer to appropriate figure number.

18. In section marginal distribution, no model fitting is shown either graphically or using KS-statistics at an appropriate significance level.

19. Some of the limitations of the study include: first, this study uses a multiple model chain which itself can lead to propagation of cascade of uncertainty based on model parameterization and initial and boundary condition of the models. Secondly, one of the assumption of development of environmental contour is environmental variables are considered as independent of time or stationary.

20. In Figure 7, environmental contours are calculated at five different points. • Authors have not mentioned the list of angles at which the calculations are performed. • Also, please mention the number of Monte Carlo simulations to derive these contours. • In x-axes of the diagram what the unit (m/0 hydro) signifies?

21. In Table 5, what CE1, ...., CE5 infer? Univariate probability?

22. Some of the words in the manuscript appears little non technical, such as "to bracket 100-year flood hazard".

23. Instead of percentage increase in flood risk in the order of 200th or 300th, which appears little unrealistic, the statistic could be presented in the form of ratio. for example, 3.84 or so.

---

## Referee Comment (RC2) · P. Schmidt-Thomé (Referee) · 30 Jun 2017

The article is very well written. It very well describes the aim of the study and its outcomes. Several methods of flood prone area mapping are described and cross-analyzed. The results presented ar very valuable, and also innovative. In other words, the article presents a valuable contribution to future flood prone area mapping, as well as current, and potential future flood prone area an extent estimation. Vulnerabilities play a key role in understanding related risks, which this article valuably focuses on, too.

---

## Author Comment (AC1) · 4 Sep 2017

We would like to greatly thank the reviewer for his support for publication.

---

## Author Response (AR1)

Dear editor,

We are grateful to you for your positive evaluation and your constructive remarks.

Below we address our response and modifications in text according to your comments.

Reviewer comments
Our response
Change in the text

- Scientific contribution of the work: Could you please put more emphasis on the scientific contribution? What is the novelty of the paper? There are some points on that in the Conclusions section, but I feel that this point should be made clearer, in particular in the Introduction.

For us, there are two main innovative aspects in our paper. First point is the modeling approach that enabled us to model the different coastal flooding process (overflowing or overtopping) and their consequences in terms of flooding at a high resolution over a spatial extent of several kilometers. To our knowledge, it is still rare to find this full process approach in the bibliography and especially at this spatial resolution and for as large computational domains.

The second point is the potential application related to the identification of the area prone to the two kind of floods. This information can be valuable for several application or multidisciplinary study around the flood risk consequences. As we now mention in text, a use of this information may be valuable for vulnerability studies, elaboration of evacuation plans or also for risk management actions.

"The solution used in our study was to link several models into a chain in order to reproduce, on the one hand, variations in mean sea level, including tides, storm surges and wave setup, by coupling a hydrodynamic model (MARS, hydrodynamical Model for Applications at Regional Scale, Lazure and Dumas, 2007) with a spectral wave model (SWAN, Simulating WAve at Nearshore, Booij et al., 1999), and on the other hand, to assess runup and overtopping volumes at the seafront, by using a NLSW model (SWASH, Zijlema et al., 2011). The chained modelling enabled us to model the different coastal flooding process (overflowing or overtopping) and consequences at a high resolution over a spatial extent of several kilometers. Overflowing and overtopping process are characterized by very different flow velocity dynamic and can cause differents impacts on structures and building. Using this modelling approach, we aim identify areas prone to one or another kind of flooding and analyze the evolutions of these two kinds of flood hazards related to local mean sea level rise. Due to the specificities of the two kind of hazards, results can be useful to vulnerability studies, to adapt people safety measure, elaborate evacuation plans, or also for risk management actions."

- Issue of breaching of seafront defences: I understood that your model chain did not include breaching, although you attempted to include the breach of the 2013 storm, correct? I think it would be important for the reader to clearly understand whether your approach includes breaching, and if it does not, what this would mean for the results.

In fact our approach do include breaching but in predetermined way (i.e. location, section, time and duration of digging are predetermined at the start of the simulation). Nevertheless in the studied case the breach was caused by the wave actions and the topographic threshold after the breach is higher than the level reached by the mean water level. So in this case, in order to simulate the breach induced floods, the simulation of the mean sea level (using the MARS2DH + SWAN) is not enough, and the second method using the SWASH model is more adapted.

We also point that the better reproduction of the observed flooded limits and flood level, is obtain when overtopping is simulated jointly with the breach.

A clarification was made in the text

- Captions of tables and of (some) figures: I believe that the captions should be made more informative. Many readers will look at the figures and tables before reading the paper in detail. Hence, I ask you to provide more information in the captions, so that a reader can understand what is given in the table or figure without trying to find the associated text. For example, I recommend to explain the scenarios, and not only to give the acronyms, such as ENC1_SLR0.

We tried to clarify and make clearer the mentioned captions

- Errors, English and style issues: The manuscript contains quite some typos. There are some French words in the figures and tables, and the English needs improvement. The first 2 bullets in the reference list are dummies. Please check carefully the complete manuscript.

We are sorry for these errors and for the style of the English of the first submitted version. We worked to improve the English and have corrected the error and French word in the second submission.

Minor comments:

- P. 2; L. 29: is it a factor of 10 (which would be 1 order of magnitude)?

Yes, we modified the text:

In some cases, the estimated value can vary by as much as an order of magnitude

- P. 1; L. 32: Here starts a very very long paragraph. I propose to split it into several paragraphs for easier reading.

Ok, we have proceed as recommended.

- Could you please check carefully all the acronym and mathematical terms, and introduce them properly? I have the feeling that several acronyms / mathematical terms are used without explaining them (e.g. WFD, APSFR, $H_s$, $T_p$) when they are used for the first time.

We completed explanation of the acronyms in text

- P. 3, L. 23: Could you please be clearer and more specific about the difference between return period of input variables and those of system responses, e.g. by giving an example? This would also help to better understand the last sentences of this paragraph where you discuss reliability methods – here I was not sure whether I completely understood what you meant.

We added examples in the text as suggested. This should make it clearer.

- P. 5; L. 1: Please give a reference for the MEDNORD model.

To our knowledge there is no reference describing the specific characteristics of the MEDNORD model. The model was elaborated during the IOWAGA project. We cited the project and we added the reference of the website of the project in the text.

- P. 5; L. 29: I had some problems in understanding which areas where used: > 20 m2 or 20-50 m2; isolated structures?

We clarified and simplified this point in text. In fact, the criterion to select the buildings was indeed structures > 20 m2

- P. 9; L. 16: Possibly related to my earlier comment: Please be more specific what you mean by "environmental contours"

The description of joint exceedance contours and environmental contours are given later in the article (cf. §3.2.1 and 3.2.2). We clarified that by referring the reader to these paragraphs.

**Response to the Reviewer #1 – Reviewer: anonymous**

1. Line 11, expected frequency of occurrence or the probability that the event will be exceeded in any year.

We clarify the text with this proposition:

Coastal flooding risks are usually defined by the intensity of flooding (spatial extent, water height, flow speed, etc. or a combination of these parameters) associated with the probability of occurrence, usually defined as the "return period".

- 2. Line 25, Expand the term SWASH model.

"the SWASH model (Simulating WAves till SHore)"

We also expand in text the MARS and SWAN model abbreviations

"(…) by coupling a hydrodynamic model (MARS, hydrodynamical Model for Applications at Regional Scale, Lazure and Dumas, 2007) with a spectral wave model (SWAN, Simulating WAve at Nearshore, Booij et al., 1999) (…)"

- 3. Line # 15 – 21, authors pointed that it does not require any assumption of dependence structure and extended to higher dimension. However, they only multivariate GPD model in their analysis. On the other hand, the marginal variables can follow any distributions, and copula modeling offers this advantage over other multivariate distributions. In addition, they also offers tail dependency through various metrics, such as,CFG, LOG or SS estimators. Extension to higher dimension is feasible using Vine or t-copulas.

We agree with the referee that copulas also offer a suitable framework to model multivariate dependency. However we believe it is somewhat more complex and also more subjective since a dependence structure exhibiting either asymptotically dependence or independence between variables is assumed beforehand. Although statistical tests exist to help selecting the best model among many, a given parametric form has to be chosen eventually even though there are usually few observation data in the region of interest (extremes).
As this is rather a personal point of view, we decided to remove the passage mentioning copulas.

In our methodology, only GPD are used to model marginal distributions. This is because GPD is the most general model for the distribution of excesses over a suitably chosen high threshold whatever the variable under study (Pickands, 1975). It is also widely used in the coastal and ocean scientific community. However, the semi-parametric model for conditional extreme values of HT04 is not limited to GPD margins. Any other marginal distribution can be used since it is transformed into Gumbel margin before fitting the dependence model.

- 4. Line 19, is the peak period unit is in 'seconds'?

Yes, we clarify this point

- 5. Some of the references are missing, such as, Stepanian et al. 2014, Lazure and Dumas, 2007. Please check others.

We are confused for this error, we include the references in the list and we checked all others

- 6. Line 25 _ 30, will land cover and land use data remain stationary since 2006 even if considering impact of urbanization?

New urbanizations or land use changes from 2006 were also updated using ortho-photographs and field observations.  This is now notified in text.

- 7. Line 6, 'Nord' is to be replaced with 'north'.

It was modified

- 8. Line 7, 'IFREMER MEDNORD' model. In summary, it would be good to give short description of the various models used in this analysis either in the Appendix or in Supplements. Further, in all cases, only abbreviation of the model names are used.
It would be nice to include full model name for the first time and use the abbreviation subsequently.

We tried to be more explicit in text in order to describe succinctly the characteristics of the used models. Particularly, we expand all the model abbreviations, we give the type of model, the source code and the resolution and for each, the main publication that supports the developments

- 9. Line 31, please provide information regarding temporal resolution of the data here. The period of data (1996-2015) used are of 20-years, but it is mentioned as 16.4 years.
Is this due to a few years of missing data?

Yes, it is due to record data interruptions and buoy maintenance.

We now precise it in text

- 10. Line 4 – 5, no autocorrelation, cross correlation analyses are shown to prove independence.

We selected this temporal criterion of 3 days based on previous research in the same area. The manuscript has been updated to mention it.

"Concerning storm dynamics in the Gulf of Lion, focusing respectively on surges and waves, Ullmann (2008) and Gervais (2012) showed that marine storms do not last longer than 3 days. We therefore decided to select the maximum Hs values per 3-day block, with a minimum of 1.5 days between peaks to make sure of their independence."

- 11. Wave angles between 60_ and 210_ were kept for the analysis. Do they represent extreme scenario?

Yes it is, given the exposure of the coastline and the wave direction during storms, only waves from the 60°-210° sector were kept for the analysis. The text was updated to be more precise.

- 12. Does threshold of the GPD model is kept as a fixed value throughout or it is considered as variables based on moving window time frame?

Once the independent events set has been derived, the threshold of the GPD for each variable is chosen based on visual tools and statistical tests. It is therefore a constant value and not a distribution parameter.

- 13. Line 14, "To finish" could be replace with "Finally" or "The next"
We modified the text.

- 14. The joint exceedance can be calculated using 'AND' and 'OR' cases. In general, the joint probability is underestimated in 'OR' case and overestimated in 'AND' case. When author(s) said 'the response variable is an underestimation' I presume calculation of return periods are performed using 'OR'-case. Refer foll. for details: Ganguli and Reddy (2013), Probabilistic assessment of flood risk using trivariate copulas. Theoretical and Applied Climatol.

Here the joint exceedance probability is calculated using 'AND' case (survival function). We thank the referee for this comment that pointed out a mistake in the manuscript. We indeed wanted to say that the joint exceedance probability of input variables (in 'AND' case) is an underestimation of the exceedance probability of the response variable.

"As underlined previously, this approach rests on the assumption that the return period of the response is equal to the return period of joint exceedance of the input variables. In reality, the joint exceedance probability of the input variables is an underestimation of the true exceedance probability of the response."

- 15. Line 26, explain the term 'instationary' in this case. In general, instationary refers to transient or quasistationary. Do authors perform nonstationary simulation in this case?

In fact, with the term in 'instationary', we mean evolutionary conditions in time representing the increasing and decreasing conditions relative to a classical storm for this area.
We modified the term in order to be clearer.

"For each scenario, a 24-hour period of evolutionary conditions (water level, waves, overtopping and propagation of inland flooding) was taken to simulate the storm conditions (including a 2h spin-up period for water level and wave conditions). This simulation time corresponds to the duration of the peak of the storm conditions regularly observed at the study site. For each scenario, the mean water level and wave dynamics at the Rank 0 limits are modelled following the shape of the 2013 storm, with concomitant water level and wave peaks at t+12h."

- 16. To analyze impact of climate change, a scenario-based analysis is performed based on earlier literature, which might carried out using older generation climate models. It would be nice to see climate change impact using finer resolution regional climate models considering future change using set of RCP scenarios after employing an appropriate bias correction scheme.

We agreed with the comment of the reviewer that it would be interesting to investigate various finer scenarios relative to climate change impact like mean sea level rise, increase of storm surge or wave peak intensity. Nevertheless, as mentioned in the text there are very

important uncertainties on the impact of climate change at local scale due to the complex ocean processes taking place in the Gibraltar Straits. We think that a more in deep analysis would be out of the scope of the paper. That is why we limited the analysis to commons scenario used for example in French National Directive on coastal flood risk.

- 17. Page 16, Line 2, "Results of the SWASH model simulations concur with these observations". Refer to appropriate figure number.

We added a new figure (FIGURE …) in order to illustrate the results

- 18. In section marginal distribution, no model fitting is shown either graphically or using KS-statistics at an appropriate significance level.

PP-plot and QQ-plot as well as p-values for KS and $\chi^2$ tests have been added for model fitting of SWL.

- 19. Some of the limitations of the study include: first, this study uses a multiple model chain which itself can lead to propagation of cascade of uncertainty based on model parameterization and initial and boundary condition of the models. Secondly, one of the assumption of development of environmental contour is environmental variables are considered as independent of time or stationary.

In the discussion, we evoke the potential error relative to the use of multiple model. For example, most of the discussion section is dedicated to the explanation of the observed errors and to the mains sources of these errors.

"Besides the errors inherent to the simulation method, there may be several reasons for the differences of a few centimetres that appeared between observations and modelling results. These include a lack of precise forcing data, the used of fixed bathymetry and potential resonance effects in the harbour that are not reproduced by the models used. Furthermore, sea levels in the northernmost pass are substantially underestimated (by 0.25 to 0.30 m). The underestimation is mainly due to the narrowness of the pass (15 to 20 m) and the potentially highly changeable bathymetry. These characteristics are the reason for the poor reproduction of water flows and levels in this sector, but do not appear to alter the results for the other sectors in the studied area. "

We also speak about the quantification of overtopping volumes which is an important field of research in the bibliography.

We agree with the second referee comment that stationarity is a fundamental assumption required by classic extreme value analyses, unless specific non-stationary methods are used to capture time dependence. In study, we assumed stationarity in the marginal distributions as well as in the dependence model. Deriving time-dependent ENC and JEC was indeed beyond the scope of the study. We completed the manuscript to discuss this point.

"Additionally, the statistical model contains uncertainties that need to be outlined. In the GPD model, a main source of uncertainties is the choice of the statistical threshold above which the distribution is fitted to the data. Estimated quantiles are indeed highly dependent on the threshold, the selection of which is sometimes difficult and often subjective despite existing statistical tools to help threshold selection (Li et al., 2012). A second source of uncertainty comes from the potential non-stationarity of the environmental variables under study. Stationarity is a fundamental

characteristic of variables required by classic extreme value analysis. Here, we assumed stationarity in the marginal distribution parameters as well as in the dependence structure of the variables Hs and SWL. Long-term trend from the SWL time series was removed before conducting the analysis but seasonal and interannual variability of SWL and Hs have not been dealt with, although this can lead to significant variations of extreme values in time (see e.g. Menéndez et al., 2009a, 2009b). However deriving time-dependent ENC and JEC (see e.g. Bender et al., 2014) was beyond the scope of this study."

- 20. In Figure 7, environmental contours are calculated at five different points. ǎA ´c
Authors have not mentioned the list of angles at which the calculations are performed.
ǎA ´c Also, please mention the number of Monte Carlo simulations to derive these contours.
ǎA´c In x-axes of the diagram what the unit (m/0 hydro) signifies?

We selected five different combinations on JEC and ENC in the upper-right quadrant which is the most important (both variables are high). We do not think it is necessary to provide the list of angles associated with the ENC combinations.

We added a precision in the text to mention the number of MC simulations used in the study: For our case study, we simulated 1 110 000 events, representing a fictitious 10 000 year period.

Thanks to the referee's comment, we modified several figures of the article that were partly displayed in French.

- 21. In Table 5, what CE1, : : :., CE5 infer? Univariate probability?

The table 5 was modified

- 22. Some of the words in the manuscript appears little non technical, such as "to bracket 100-year flood hazard".

We update the manuscript changing the term.

- 23. Instead of percentage increase in flood risk in the order of 200th or 300th, which appears little unrealistic, the statistic could be presented in the form of ratio. for example, 3.84 or so.

Thank you for this suggestion, we now present the results in the form of ratio.

[revised manuscript text omitted]

---

## Author Response (AR2)

Reviewer comment

Response to the rewiever

Correction in text

5    1. Page 3, line 9, why SWASH model is kept within brackets after NLSW model? Is it under the same family as NLSW?

Yes SWASH model is part of the Non Linear Shallow Water model family.

2. Page 3, line 11, different in place of differents.

Correction was made

3. In Page 4, line 9, did you really used any reliability methods to assess the coastal flood risks? I don't find this in the paper. Probably, this is a better idea to discuss these issues in the discussion section instead of right in the beginning. There are various ways discussed in the past, for example, design life level (Rootzén and Katz, 2013), to quantify the probability of exceeding a fixed threshold during the design life of a project, replacing the commonly used concept of average return period

15   with reliability (Read and Vogel, 2015), and a risk-based decision-making approach integrating the concept of expected regret (Rosner et al., 2014).

The approach of Huseby et al. (2013) used in section 3.2.2 is similar to the classic inverse first-order reliability method (iFORM) commonly used in structural design. We modified the introduction to clarify this point, as well as the beginning of section 3.2.2.

20   Introduction

In the related field of structural engineering (design of coastal defences, offshore renewable energy systems, etc.), it is usual to refer to the environmental contour concept in conjunction with the inverse first-order reliability method (iFORM) (Winterstein et al., 1993; Jonathan and Ewans, 2013 ; Huseby et al., 2013; Huseby et al., 2015). Such an approach focuses on extreme system responses rather than on the combinations of extreme environmental loads. The idea is to identify a set of

25   design environmental loads (e.g. contours in 2D) within which extreme responses with a given return period should lie. In other words, given the failure probability (or return period) of the system response, the objective is to identify what kind of restrictions this imposes on possible designs. This approach is rarely used to assess coastal flooding risks. In this paper, it will be compared with the more classic method, where in choosing the scenarios it is assumed that the return period defined from the input variables corresponds to the return period of the system response, in order to analyse the differences that arise

30   from the methods used to define the scenarios.

Section 3.2.2

A second approach involves using environmental contours (written here as enc), which are commonly used in offshore structural engineering (e.g. Huseby et al., 2013, 2015; Jonathan and Ewans, 2013). These contours are defined within the input variables space but are based on the probability of exceedance of the response variable. These methods rest on an

approximation of the limit state curve and are independent from the model. A classical way of defining such environmental contours is to use the inverse first-order reliability method (iFORM) (Winterstein et al., 1993). Here we preferred to use the approach developed by Huseby et al. (2013, 2015) as it overcomes some limitations of the iFORM and it is especially suited to Monte Carlo simulated datasets. An environmental contour defined in this way is an (x,y) contour in the space (SWL, Hs), outlining a convex inner surface. (...)

In addition, following the editor's suggestion, we modified the discussion section as follows:

A second source of uncertainty comes from the potential non-stationarity of the environmental variables under study. Stationarity is a fundamental characteristic of variables required by classic extreme values analysis. Here, we assumed stationarity in the marginal distribution parameters as well as in the dependence structure of the variables Hs and SWL. Long-term trend from the SWL time series was removed before conducting the analysis but seasonal and interannual variability of SWL and Hs have not been dealt with, although this can lead to significant variations of extreme values in time (see e.g. Menéndez et al., 2009a, 2009b). However deriving time-dependent ENC and JEC (see e.g. Bender et al., 2014) was beyond the scope of this study. To go one step further, this issue of potential non-stationarity of the environmental variables questions the relevance of the classic concept of average return period to characterize the risk of coastal flooding. Indeed, the average return period provides information about the probability of exceeding a threshold in any given year. It does not inform about the cumulative risk over a given period of time, which is of interest when it comes to the design of coastal defences for example (cf the design life period of the structure and the concept of reliability, see e.g. Read and Vogel, 2015; Rootzen and Katz, 2013). This discussion also leads us to question the general framework one uses to assess the risk of coastal flooding. A risk-based approach, starting from the end-users needs rather than from a fully scientific analysis only based on physical and statistical considerations, might be better suited to take into account the planning horizon of the study (the design life period in the case of structure design) as well as various aspects such as risk perception (Idier et al., 2013) or economic factors (Rosner et al., 2014).

4. Page 15, line 1, what is H&T04 method?

It refers to the Heffernan and Tawn (2004) method. This was clarified in text.

5. Page 15, lines 16 ~ 20, regarding SLR projection at different RCP levels, a recent paper by (Kopp et al., 2014) is worth mentioning in the discussion section of the manuscript though authors have not used them in their study.

Thank you to the reviewer for this suggestion. We added the reference in text.

[revised manuscript text omitted]